# Interferences caused by the biogeochemical methane cycle in peats during the assessment of abandoned oil wells

Sebastian F. A. Jordan[1]*, Stefan Schloemer[1], Martin Krüger[1], Tanja Heffner[2], Marcus A. Horn[2], Martin Blumenberg[1]*

[1]Federal Institute for Geosciences and Natural Resources (BGR), Hannover, Stilleweg 2, 30655, Germany
[2]Leibniz Universität Hannover, Institute for Microbiology, Herrenhäuser Str. 2, Hannover, 30419, Germany

*Correspondence to: Sebastian F. A. Jordan (Sebastian.Jordan@bgr.de), Martin Blumenberg (Martin.Blumenberg@bgr.de)*

**Abstract.**

In the global effort to reduce anthropogenic methane emissions, the millions of abandoned oil and gas wells are suspected to be prominent but so far often overlooked methane sources. Recent studies have highlighted the hundreds of thousands of undocumented abandoned wells in North America as major methane sources, sometimes emitting up to several tons of methane per year. In Germany, approximately 20,000 abandoned wells have been described, which are well documented, and the data is publically available. Here we present a methodological approach to assess particularly methane emissions from cut and buried abandoned wells, which are typical for Germany. We sampled eight oil wells in a peat-rich environment with four wells in a forest, three wells in an active peat extraction site, and one well on a meadow. All three areas are underlain by peat. At each site, we sampled a 30 x 30 m grid and a corresponding 20 x 20 m reference grid. Three of the eight wells and reference sites exhibited net methane emissions. In each case, the reference sites emitted more methane than the respective well site with the highest net emission (~110 nmol $CH_4$ $m^{-2}$ $s^{-1}$) observed at one of these reference sites. All methane-emitting sites were located within the active peat extraction area. Detailed soil gas characterization revealed no methane, ethane, and propane ratio typical for reservoir gas, but instead showed a typical biogenic composition and isotopic signature (mean $\delta^{13}$C-$CH_4$ –63‰). Thus, the escaping methane did not originate from the abandoned wells or the associated oil reservoir. Furthermore, isotopic signatures of methane and carbon dioxide suggest that the methane from the peat extraction site was produced by acetoclastic methanogens, whereas the methane at the meadow site was produced by hydrogenotrophic methanogens. However, our genetic analysis showed that both types of methanogens were present at both sites, suggesting that other factors control the dominant methane production pathway. Subsequent molecular biological studies confirmed that aerobic methanotrophic bacteria were also important and that their relative abundance was highest at the peat extraction site. Furthermore, the composition of the methanotrophic community varied between sites and depths. The aerobic methane oxidation rates were highest at the peat extraction site potentially oxidizing a multiple of the emitted methane, thus likely providing an effective microbial methane filter.

For the assessment of potential leakage from cut and buried abandoned wells, our results highlight the need to combine methane emissions with soil gas characterization in comparison to a suitable reference site. A monitoring that relies exclusively on methane emissions may result in erroneous classification of naturally occurring emissions as well integrity failure.

# 1 Introduction

Methane is one of the key greenhouse gases contributing to climate change. It has, however, a particular role in climate change mitigation, too, as its atmospheric lifetime is especially short (Saunois et al., 2020). This makes methane the prominent political
target, because emission reduction may quickly result in decreasing atmospheric concentration and, thus, climate effect. In total 156 countries participate in the global methane pledge aiming to reduce global methane emissions by 30% until 2030
(IEA, 2024).

The fossil fuel sector is the second largest anthropogenic methane source. Methane emissions from this sector not only include

emissions from active production but also leakage from millions of abandoned oil and gas wells, due to well integrity failure as recent studies found, e.g., in the USA and Canada (Samano et al. 2022, Williams et al. 2021). The permanently increasing
number of global abandoned oil and gas infrastructure is a rising problem (Bowman et al. 2023, Williams et al. 2021, Riddick et al. 2020), which will intensify in the future during the transition to renewable energy sources. Abandonment procedures
today depend on national regulations and are now often similar although they have differed strongly in the past. However, the greatest impact on the country's abandoned well situation has probably been the extent to which these regulations were
properly enforced. Some countries are additionally struggling with the situation of undocumented or orphaned wells (Boutot et al. 2022). Different well abandonment practices throughout history have meant that in some cases only the well head has
been closed and everything has been left in place while the well casing was unplugged (Pekney et al. 2018, Williams et al. 2020), in others an open bore hole was left in the ground (Pekney et al. 2018, Lebel et al. 2020), or the wells were properly
plugged, cut and the remains buried (Davies et al. 2014, Schout et al. 2019, Cahill et al. 2023). Thus, authorities and scientists try to identify particular high emitters (Bowman et al 2023) or wells with a high risk of integrity failure (Cahill and Samano
2022) to maximize economic and environmental benefits (Kang et al. 2021) as financial resources for proper well decommission are limited (Raimi et al. 2021, Agerton et al. 2023). In Germany for example, first regulations date back to 1904
and were refined every few decades to the last update of 2006 (von Georne et al. 2010). All kinds of wells (exploration, production, appraisal and injection wells) are generally decommissioned and buried (Landesamt für Bergbau, Energie und
Geologie (LBEG), 1998).

It is therefore not possible to detect methane emissions from such wells using the same methods as for wells with visible

surface installations, such which are partly found in the US and Canada (Williams et al. 2021, Lebel et al. 2020). For cut and buried wells (e.g., in Germany, the Netherlands, and UK), single measurements atop the wells location are insufficient (Schout
et al. 2019). In this case, upwards migrating natural gas can be subject to several physical and biogeochemical processes, e.g., microbial oxidation is able to alter concentrations and even isotopic composition (Whiticar 2020). Leaking gas can even
migrate away from the wells location (Dennis et al 2022, Forde et al. 2019a), disperse through the soil and potentially be oxidized by methanotrophic microorganisms on its way to the atmosphere (Forde et al. 2022). Thus, false negative results on
the well integrity would be obtained. In addition, biogenic methane can be microbially produced in shallow anoxic soils by methanogenesis. Thereby, organic carbon degradation facilitated via a complex network of trophically linked microorganisms
(e.g., intermediary ecosystem metabolism, Drake et al. 2009) ultimately resulting in methane production when alternative electron acceptors except for carbon dioxide are depleted (Whiticar 2020). Methanogenesis is mainly carried out by three types
of anaerobic archaea in more than 30 genera: 1) acetoclastic methanogens converting acetate to methane and carbon dioxide, 2) hydrogenotrophic methanogens, reducing carbon dioxide to methane with hydrogen, and 3) methylotrophic methanogens
disproportionating methyl groups to methane and carbon dioxide (Liu and Whitman, 2008). Although most methanogenic species are hydrogenotrophs, two-thirds of biologically produced methane is derived from acetate (Liu and Whitman, 2008).
Combining isotopic composition of methane and the relation of methane to the sum of ethane and propane is an often-used method to distinguish natural gas (commonly thermogenic) from biogenic methane (Whiticar 2020). However, methane can
further be oxidized by anaerobic and aerobic methanotrophs to carbon dioxide along its way to the atmosphere, which shifts the isotopic composition, adding more complexity. In case of organic rich soils or soils with a high groundwater table, methane
production can outweigh its consumption leading to substantial methane emissions (Le Mer and Roger 2001, Lai 2009). To put this into perspective, upland forests are known to act as methane sink taking up to $\sim$–4 nmol m$^{-2}$ s$^{-1}$ methane from the
atmosphere, whereas natural wetlands emit up to $\sim$600 nmol CH$_4$ m$^{-2}$ s$^{-1}$, which can be topped by rice paddy fields with over 2000 nmol CH$_4$ m$^{-2}$ s$^{-1}$ (Oertel et al. 2016). In general, these processes are taking place in the active zone of the soil in general
but especially environments with biogenic methane generation are prone to generate false positive well leakage classification. An example for such complex environments are wetlands and peat rich areas that are associated with $\sim$ 2700 abandoned wells
in Germany (mainly in Northern Germany), translating to roughly 15% of all abandoned German wells ($\sim$24,000; NIBIS® Kartenserver 2014b, Wittnebel et. al., 2023). Thus, these areas act as an ideal test ground for method testing. Peat rich areas
are biogeochemical complex and are defined as former raised/ombrotrophic bogs, rich fens and other types of peat accumulating wetlands. In the pristine ecosystems, the vegetation is taking up carbon dioxide from the atmosphere and
producing biomass. Peat accumulates as plant litter and is only partially decomposed due to oxygen limitation (Turetsky et al., 2014; Frolking et al., 2006) below the partially aerated and very thin vadose zone. However, most raised bogs in Central
Europe were drained in the past for agricultural use, forest cultivation, and peat extraction for fuel or horticultural purposes (Pfadenhauer and Klötzli, 1996; Laine et al., 2013). After drainage, most of these wetlands change from net carbon sinks to
net carbon sources (Frolking et al., 2006). This is due to the remineralization of once stored organic matter to ultimately carbon dioxide (Abdalla et al., 2016). Then again, methane emission decreases drastically as the aerated soils enable aerobic methane
oxidation to CO$_2$ and methanogenesis is restricted to deeper layers (Sundh et al., 1994; Abdalla et al., 2016). Nonetheless, the greenhouse gas balance changes with drainage and differs depending on land use (Abdalla et al., 2016). Methane emissions
are thought to stop altogether in peatlands used for forestry or agriculture (Abdalla et al., 2016 and references therein). However, previous studies point towards substantial methane emissions from ditches, which are draining the peats and can
even reach the magnitude of emissions from virgin peatlands (Sundh et al., 2000). The extraction of peat results in an accelerated carbon loss and increased greenhouse gas emissions as peat decomposition associated with end use comprises the
majority of total emissions (e.g., combustion and use in horticultures; Cleary et al., 2005). In this complexity of methane and

carbon dioxide related biogeochemical processes in soils in general, one has to look closely to delicately allocate methane emission to natural or anthropogenic (e.g., abandoned wells) sources.

Worldwide only very few countries, e.g., the USA and Canada (Bowman et al., 2023) include emission from abandoned wells in their yearly greenhouse gas inventory. In a BGR project "leakage assessment of buried wells in Germany", we aim to fill this knowledge gap for Germany by studying a representative sub-set of abandoned wells. We use the term "abandoned well" here to refer to a former oil or gas well in Germany that has been decommissioned and buried in accordance with the guidelines in force at the time (von Georne et al. 2010). This includes plugging and backfilling of the well, cutting, and removing of the shallow casings, and reconditioning of the area (e.g. for agricultural use).

Here, we present a first detailed study of eight wells in a complex methane rich setting in Northern Germany. Environments with high in-situ biogenic methane generation might lead to a false positive well leakage classification based on surface emission measurements, if the methane source (shallow biogenic vs thermogenic natural gas) is not correctly determined. We present our principal methodological approach, a combination of geochemical and microbial techniques, to evaluate methane emissions from cut and buried abandoned wells. In this paper, we focus on the results from this small study area, including overall methane emissions and identifying the source of the methane, and thus allocate the emissions to the abandoned wells or in-situ methanogenic processes. In addition, the microbiological methods enabled us to quantify the methane oxidation potential of the soil, i.e., the methanotrophic methane filter function, and identify key organisms feeding on the soil methane.

## 2 Methods

### 2.1 Study site

The sampling and field measurements were conducted in March and April 2022 near Steimbke (Lower Saxony, Northern Germany), an area with ongoing and historical industrial peat production. Additional samples were taken in April 2023 from the peat extraction site and November 2023 from reference sites. Three oil fields were located around Steimbke. From these three, we focused on field Steimbke-Nord. Data including location, depth, date of drilling, etc. of wells related to this oil field as well as the other ~20,000 wells in Germany (abandoned, producing, and exploration) and data on the oil and gas fields are publicly available via NIBIS map server (NIBIS® Kartenserver 2014a, 2014b). We used this database to locate about 200 wells in the vicinity of Steimbke-Nord including 159 abandoned production wells. The oil-bearing geological horizons were located in the Malm and Dogger (both Jurassic) in 500 to 700 m depth covering an area of about 1.5 km$^2$. The wells were drilled between 1942 and 1950 and are typically 570 to 695 m deep. In total 3 x 10$^8$ t oil (and 2.9 x 10$^9$ m$^3$ oil associated natural gas) were produced until 1964 (https://nibis.lbeg.de/cardomap3/?permalink=WeOGYg3, accessed 03.05.2024). We studied and sampled eight abandoned wells, each with respective reference measurements (Figure 1, Table 1). To investigate methane emissions related to abandoned onshore wells eight cut und buried wells in the south-eastern part of this oil field covering an area of ~ 0.2 km² (Figure 1) were targeted. The eight wells are situated in three different land use types. Three wells (R-WA

272, R-WA 254, R-WA 264) are located in the western part of the area where active peat mining is ongoing with the bare peat directly at the surface (from here on "Peat" sites). Before the peat extraction in the active area began, the Peat site was also an agricultural meadow that was probably temporarily grazed and regularly fertilized with manure like the meadow at well site R-WA 275, ~ 350 m to the east (from here on "Meadow" site). Two of the four wells from the forest area (dominated by birch trees and pines) are located between the active Peat site and the Meadow (R-WA 273, R-WA 274), the remaining two in a larger forested area ~ 225 m to the north and northeast, respectively (from here on "Forest" sites). In case of the Forest and Meadow sites, the top soil above the peat layer was sampled, whereas at the Peat sites the peat was sampled directly. Regarding pH of the Peat site, Welpelo et al. (2024) published a pH of ~ 3.5 for a nearby rewetted part of the peat extraction area, about 2.5 km away as well as additional physicochemical parameters. Residues from the drilling and/or production were only human eye visible in the forest area. Here, cement residues likely from the rig cellar or associated infrastructure, sand from the backfill procedure, and small depressions were signs of former activity. No residues of the former well itself, like wellheads, old horsehead pumps, or any kind of piping were visible. All sample sites are situated in a peat rich area and the majority of sites include about 1.0 m or more raised-bog peat either below the topsoil (Forest, Meadow) or as bare peat at the Peat site (https://nibis.lbeg.de/cardomap3/?permalink=1baQ8yzX, accessed 03.05.2024). Peat depth in this area was taken from a geological exploration in 1983. An exemplary soil profile is shown in Fig. 2d, this profile was drilled near our peat reference site (~50 m west). These profiles show a peat thickness of ~1.9 m up to ~2.6 m for the peat sites with about 1 m and more already extracted since ~2017. For sites R-WA 273, R-WA 274, R-WA 275 the state agency (https://nibis.lbeg.de/cardomap3/?permalink=1uIMU2yt, accessed 03.05.2024) estimated a peat thickness of more than 2 m. However, for sites R-WA 211 and R-WA 209 peat was confirmed with more than 30 cm depth but its entire thickness is unknown.

**Table 1:** Overview of surveyed well locations and selected meta data. All wells were used for oil production in the past. *peat is present in all areas (Peat = active peat extraction site)

| name | short name | north | east | drilling completed | depth (m) | area* |
|---|---|---|---|---|---|---|
| Rodewald-WA 211 | R-WA 211 | 5836503 | 32525924 | 26.10.1942 | 635.5 | Forest |
| Rodewald-WA 209 | R-WA 209 | 5836399 | 32526148 | 27.08.1942 | 570.5 | Forest |
| Rodewald-WA 273 | R-WA 273 | 5836338 | 32525761 | 03.08.1950 | 682.7 | Forest |
| Rodewald-WA 274 | R-WA 274 | 5836299 | 32525835 | 04.07.1950 | 680 | Forest |
| Rodewald-WA 275 | R-WA 275 | 5836302 | 32525931 | 21.07.1950 | 670 | Meadow |
| Rodewald-WA 272 | R-WA 272 | 5836374 | 32525686 | 15.06.1950 | 700 | Peat |
| Rodewald-WA 254 | R-WA 254 | 5836366 | 32525498 | 15.12.1948 | 695 | Peat |
| Rodewald-WA 264 | R-WA 264 | 5836323 | 32525566 | 03.06.1950 | 660 | Peat |

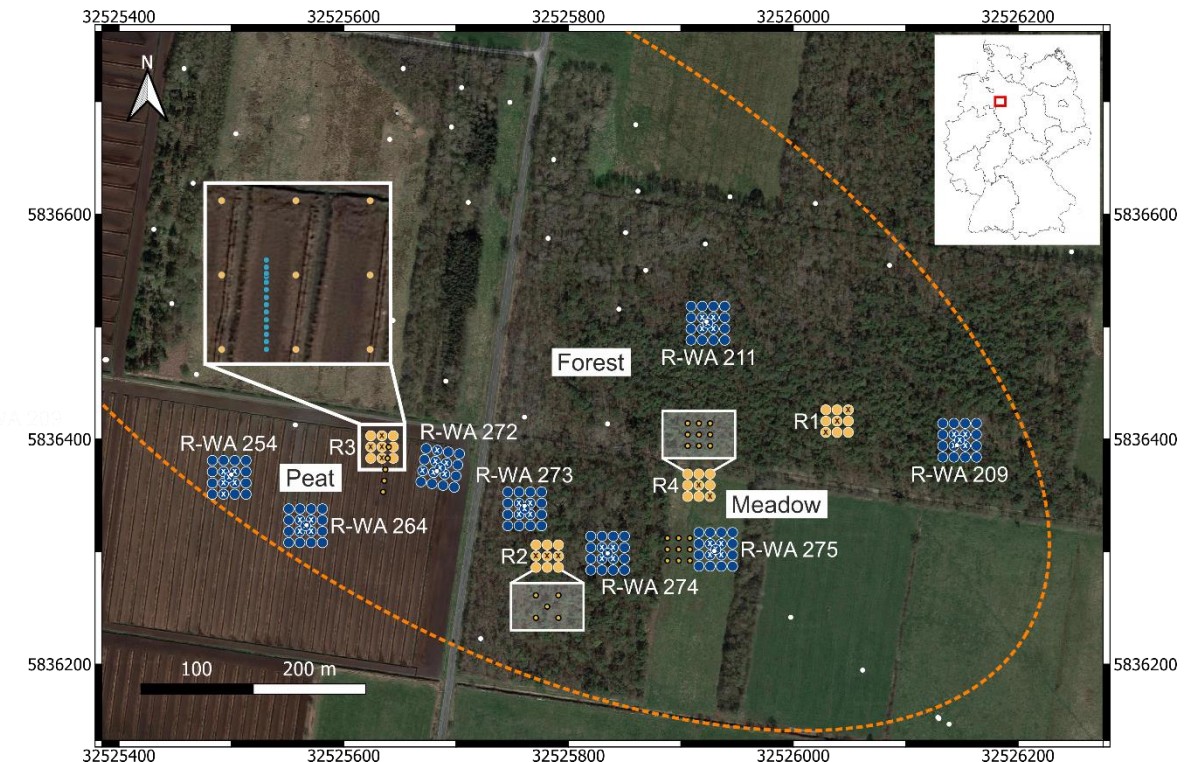

**Figure 1:** Overview of the study site in Steimbke with the well sites and reference site measuring grids each with 17 and 9 measuring points, respectively. Abandoned wells are depicted in white dots and those studied here are named (e.g., R-WA 211 etc.). The orange dotted line outlines the rough dimensions of the oil field Steimbke-Nord. Coordinates are stated in UTM-32U (WGS84) with easting and northing planar coordinates in meter. Blue indicates the well site emission ($CH_4$ and $CO_2$) measuring grid whereas orange indicates the reference site emission ($CH_4$ and $CO_2$) measuring grids with the positions for soil gas sampling marked as white or black "x", respectively. The left inlet depicts a transect with ~1 m distance between the measuring points to assess spatial variations in an area without a well. Additional soil gas sampling points are depicted as orange circles and are shown in part in a white box for better visibility. The areas compared in this study ("Peat", "Forest", "Meadow") are also marked. The map was created using QGIS (v.3.22.3) and © Google Earth satellite images from 2015 as background.

## 2.2 Sampling method and grid

We studied well and reference sites for methane (and $CO_2$) emission (positive and negative), soil gas composition and microbial communities (Figure 2c). The reference sites were placed in a distance of 15–150 m to any studied well on the same terrain. The position of the wells was extracted from the NIBIS® MAPSERVER (NIBIS® Kartenserver 2014), and a handheld GPS device (Garmin, etrex Vista Hcx) was used to navigate in the field. Due to the burial of abandoned wells in the working area, our study relied on the coordinates of the wells. Discussion with the LBEG, the local public as well as indications (e.g., color changes, remnants of roads/pathways) from recent and historical Google Maps images supported the correctness of the well positions.

The central measuring point was placed directly above the well. We positioned the other 16 measuring points around the well

pointing north with the help of two measuring tapes and a compass. The distance between these 16 points was 10 m from point
to point aiming at a broad coverage of potential methane emission areas above the buried wells. In total, the well site grid

covered an area of 30 x 30 m and 17 measuring points (Figure 2a). Soil gas samples were taken in the central five positions of
the well as indicated in Fig. 2a. Soil samples for microbial analyses were usually taken at three positions starting at the center

towards one of the corners. In case of high methane emissions, additional soil gas and microbial samples were taken at the
respective spots.

For these eight wells, we established four different reference sites R1 to R4 (Figure 1). The reference grids consisted of nine
measuring points covering an area of 20 x 20 m (Figure 2b). Measuring reference grids is necessary to determine and account

for potential natural background variations for each abandoned well. Reference grids were typically located in a distance of
15–150 m from the well site and on similar soil conditions and vegetation, and were investigated immediately after the

measurement of the well grid. The reference site R4 for the abandoned well on the Meadow was measured once and the two
reference sites for the wells in the Forest area (R1 and R2) were each measured twice on consecutive days (Table 2). The single

reference site for the three wells in the Peat area (R3) was thus surveyed three times within one week. Three soil gas samples
were usually taken in a diagonal pattern and the soil sample for microbial analysis in the center of the grid (Figure 2b). To

estimate the general emission's spatial variability in the area, we sampled a transect through a point with high emission at the
Peat reference site. The measuring points along the 12 m transect were 1 m apart.

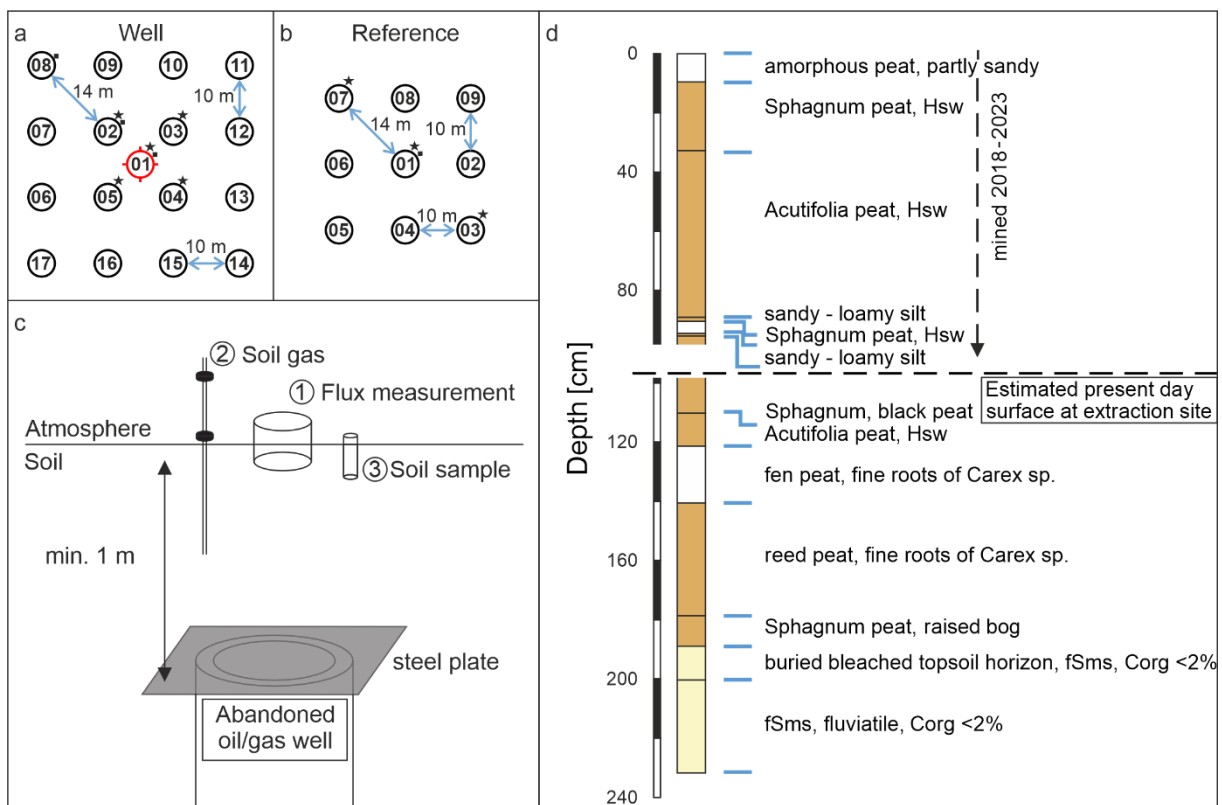

**Figure 2:** Sampling scheme for emission measurements ($CH_4$ and $CO_2$) for well (a) and close-by reference sites (b), both with likely similar biogeochemistry and vegetation, as well as a schematic display of a buried abandoned well (c, not by scale). Additional soil gas samples (stars) and soil samples for microbial analysis and methane oxidation rates determination (squares) were taken at the marked positions. The shift of the symbols towards the upper right was made for graphical reasons. Samples were directly taken on the numbered positions. The well position is marked in red. d) A simplified profile of a pedological well (#54315, source LBEG) drilled in 1983 before peat extraction initiated. Coordinates RW: 32525578, HW: 5836405 (EPSG 4647), close (50 m west) to the reference grid in the peat extraction site. fSms: medium sandy fine sand, Hsw: diffuse or in nests enriched with unconsolidated sesquioxides. Link to map https://nibis.lbeg.de/cardomap3/?permalink=2RfGItuF.

## 2.3 Methane and carbon dioxide emission

Methane emissions from the soil surface into the atmosphere were measured with an optical feedback – cavity enhanced absorption spectroscopy trace gas analyzer (LI-COR 7810) coupled to a portable hydraulic chamber (LI-COR smart chamber) following the closed chamber principal. The measurement was conducted as instructed by the manufacturer, which is described in the following: First, defined plastic collars with a diameter of 20.3 cm and height of 12.4 cm (outside diameter 8.4", height 4.5") were positioned at each of the measuring points and were pushed few centimeters into the soil to guarantee a complete closure of the smart chamber with the underlying soil profile. Since the exact penetration depths of the collars were needed for the calculation of fluxes (dead volume of the ring) each insertion depth were measured individually.

After both devices, analyzer and chamber, reached operation modus, a startup measurement (triplicate) was conducted to ensure stable condition of the instrument. Each grid position was sampled in triplicates at least 1 h after placing the respective
collar. The chamber stayed closed for the time of one measurement (120 s) to record continuously (1 Hz) the change in methane and carbon dioxide concentrations in the loop headspace, which was open to the soil surface. Gas fluxes were computed after
a dead band time of 40 s after chamber closing, applying a linear regression of the concentration data for each singular measurement, subsequently averaging the triplicate measurements. Between measurements, the chamber stayed open for 60 s
to enable equilibration with atmospheric $CH_4$ and $CO_2$ concentrations. Examples for such measurements and their $r^2$ are shown in the supplements (Supplement S1) and the standard deviation of the triplicate measurements is tabulated in supplementary
data (Table S2).

Additional measurements at each site included soil moisture (SWC) and bulk conductivity measurements (EC) using a Stevens

HydraProbe sensor with 6 cm long measuring rods. The sensor was not particularly calibrated for the organic (peat) rich soils at the study site and used the default "sand" settings for data evaluation. Thus, reported SWC and EC data in Table S2 are only
indicative data. Since the short rods measured the temperature effectively directly below the soil surface with all potential bias to solar radiation, we applied an additional 25 cm long temperature probe (Omega, Type E) to better constrain soil
temperatures. In addition, accompanying weather data, measured with a handheld device (e.g., temperature, wind speed and humidity) can be found in Table S8.

**2.4 Soil gas sampling and compositional analysis**

Gas samples were acquired using soil gas probes. The probes are made of stainless steel with an outer diameter of 6 mm and inner diameter of 3 mm with a total length of 1.5 m. To prevent the probes from becoming blocked while pushing into the
ground, a pin has been attached to the front of the probe. This pin remains in the ground after the desired depth is reached and the probe was lifted by a few cm. The lances are usually driven into ground with a moveable anvil. However, they could easily
be pushed into the maximum depth of 1 m at study sites with soft unconsolidated soils. Locations sampled are indicated in Fig. 3 and wherever methane emission was detected. Due to the shallow ground water table, the probes often had to be lifted close
to the surface to be able to sample the gas phase of the vadose zone, thus giving an approximate indication of the actual water level (sampling depths are listed in Table S1). A septum port is attached to the end of the probe, which allows sampling with
a syringe. Before sampling, the dead volume of the soil gas probe was flushed twice with soil gas immediately after placement with a 20 mL syringe and then rested at least for 1 h to equilibrate. Afterwards 20 mL soil gas was extracted and stored in
crimped vials pre-filled with saturated NaCl as sealing solution. Vials were stored upside down for a maximum of about two weeks before further gas analysis in the laboratory.
Stored gas samples were analyzed in the lab using a gas chromatograph (GC) Trace 1310 GC (Thermo Fischer Scientific, USA) equipped with a heated valve system and column switching. One milliliter of sample was then injected into the sample
loops. The individual components were quantified in parallel on three channels. On channel 1, pre-separation of hydrocarbons

($C_1$ through $C_6$) from a 500 µL sample was performed on a non-polar polysiloxane polymer column (Restek MX-1, 15 m, 0.28

mm ID, film thickness 3 µm). Molecular weight components $>C_7$ were back-flushed. Full separation was performed on the

main 50 m $Al_2O_3$ capillary column (0.32 mm ID, film thickness 5 µm). Both columns were operated non-isothermally starting

at 30°C and ending at 180°C. All components were detected on a Flame Ionization Detector (FID) with helium (He) as carrier

gas. On channel 2, the sample was injected via a 500 µl sample loop. $CO_2$ was separated from other components by a pre-

column (30 m Hayesep Q, 0.53 mm ID, film thickness 20 µm) and directly detected after bypassing the Molsieve column on

the thermal conductivity detector (TCD). All other components (Ne, $H_2$, Ar, $O_2$, $N_2$, $CH_4$, and CO) were chromatographically

separated on the main analytical Molsieve column (80 m 5 Å, 0.53 mm ID, film thickness 50 µm). Carrier gas on this channel

was He. For better sensitivity for helium and hydrogen, these compounds were analyzed on a channel 3 with argon as carrier

gas. The sample loop used had a volume of 125 µl. $CO_2$ and higher molecular weight carbon-components were retained and

back-flushed on a packed pre-column (2 m Hayesep Q, mesh 100/120, 1 mm ID). Separation of He, Ne, $H_2$, $O_2$, and $N_2$

components was performed on a 5 Å packed molecular sieve column (3 m, mesh 80/100, 1 mm ID) and subsequently detected

on a TCD.


## 2.5 Isotopic analysis of methane and carbon dioxide

For samples with concentrations (>200 ppm), carbon isotope signatures of $CH_4$ ($\delta^{13}$C-$CH_4$) and $CO_2$ ($\delta^{13}$C-$CO_2$) were

determined after injecting into a continuous flow GC-IRMS system (Agilent GC coupled to a Thermo Fisher Scientific MAT

253 via a GC-Combustion interface II/III). The different compounds were separated on a 25 m Porapak column and methane

was combusted to $CO_2$ at a temperature of 960°C. Low concentration samples (2 – 200 ppm $CH_4$) were measured applying a

cryo-focusing with liquid nitrogen of methane on a 1 m 1/16 packed column installed in an Agilent 6890 GC likewise coupled

to a Thermo Fisher Scientific MAT 253 via a GC-Combustion interface II/III. Deuterium isotope signatures of methane ($\delta^2$H–

$CH_4$) were determined by a similar GC-IRMS system (Trace GC and Isolink/ConFlow IV coupled to a MAT 253) if methane

concentrations were above 2000 ppm. Methane was reduced to molecular $H_2$ at a temperature of 1420°C. The reproducibility

for $\delta^{13}$C is ± 0.3‰ and for $\delta^2$H–$CH_4$ ± 3‰. $^{13}$C/$^{12}$C and $^2$H/$^1$H ratios are presented in the standard δ-notation versus the reference

standards Pee Dee Belemnite (VPDB) and Standard Mean Ocean Water (VSMOW), respectively (Coplen, 2011).


## 2.6 Methane oxidation rates

In the field, shallow soil samples (down to 20 cm) were obtained using a stainless steel push core with an inner Plexiglas liner.

Exact coordinates and sampling depth are listed in the supplementary data (Table S4). Deeper samples (40–100 cm) were

retrieved with the help of an Edelman auger as a 20 cm composite sample. Samples were kept, transported, and stored at 4–

7°C until further processing. As next step, samples were homogenized and 5 g subsamples were collected and stored at –20°C

for DNA extraction. For determination of potential aerobic methane oxidation rates (MOx) each sample was divided into seven

aerobic incubations (100 mL vials), with ~10 g homogenized soil sample in each. Three parallels were incubated with 1% methane in the headspace, four without methane with one being autoclaved prior to incubation.

Headspace methane and carbon dioxide concentration were determined regularly with a 610C gas chromatograph (SRI Instruments Europe GmbH, Bad Honnef, Germany) equipped with a flame ionization detector (FID). At the end of the incubations, bottles with active soil samples were subsampled for DNA extraction again (s. section DNA extraction) and then the remaining sample was dried at 80°C to calculate soil water content. In the end, methane oxidation was calculated as the slope of the declining methane concentration in µmol per incubation over time in a linear section of the graph. Subsequently, it was then accounted for the dry weight in case of MOx dry and the wet weight for MOx wet. Finally, to compare it to methane emissions MOx wet was multiplied by the respective soil density and a volume of 0.2 m$^3$, because 20 cm was the maximal depth of a composite sample.

## 2.7 DNA extraction

DNA was extracted from soil samples (~0.5 g) using the FastDNA SPIN kit for soil (MP Biomedicals, Illkirch, France). The extraction followed manufacturer's instructions with modifications as previously described Webster et al. (2003): (1) the addition of 200 µg of poly(adenylic acid) (Roche Diagnostics International Ltd., Rotkreuz, Switzerland) prior to bead beating; (2) two bead beating steps of 45 s at 6.5 m s$^{-1}$ were performed on a FastPrep-24 system (MP Biomedicals); and (3) DNA was eluted in TE-buffer and quantified with the Quantifluor dsDNA chemistry using a Quantus fluorometer (Promega GmbH, Walldorf, Germany).

## 2.8 Sequencing bacterial and archaeal community via 16S rRNA genes

Following DNA extraction, samples were sequenced by Microsynth AG (Balgach, Switzerland) using MiSeq Illumina technology for microbial community analysis. Both bacteria and archaea were sequenced from the same DNA extractions and analyzed separately by targeting the 16S rRNA gene. For bacteria primer pair 515F / 806R (GTG CCA GCM GCC GCG GTAA; GG ACT ACH VGG GTW TCT AAT; Caporaso et al. 2011) and for archaea 340F / ARCH806R (CCC TAY GGG GYG CAS CAG; GGA CTA CVS GGG TAT CTA AT; Takai and Horikoshi 2000; Gantner et al. 2011) were used. Sequences will be deposited in the European Nucleotide Archive (ENA) and the accession number will be published in the final version of the manuscript. Sequences were processed following a bioinformatics pipeline (USEARCH, Edgar 2010; Cutadapt, Martin 2011; MOTHUR, Schloss et al. 2009) previously described by Dohrmann and Krüger (2023). Thereby zero-radius OTUs (ZOTUs) ( are generated from OTUs using the UNOISE algorithm, which enable higher resolution with the goal to report all correct biological sequences (Edgar 2016). Potentially methanotrophic ZOUTs were identified according to the *pmoA* database taxonomy (Yang et al., 2016) and known methanotrophic genera (Knief, 2015, 2019 and references therein). Relative

abundances of a methanotrophic genus or family were calculated as the share of all methanotrophic genera or families in the respective sample pool.

## 2.9 Quantification of methane oxidizing bacteria by *pmoA*-gene targeted quantitative PCR

Using quantitative PCR (qPCR) assays to targeting both, general bacterial 16S rRNA gene and the *pmoA* gene encoding for the ß subunit of the particulate methane monooxygenase expressed by methane oxidizing bacteria (MOB), we were able to
determine the methanotrophic abundances.

The qPCR targeting the 16S rRNA gene (primer pair 341F/ 805R; forward: 5′-GTGCCAGCMGCCGCGGTAA-3′, reverse:
5′-GGACTACHVGGGTWTCTAAT-3′) was performed as described previously (Hedrich et al., 2016). The *pmoA* gene targeting qPCR (primer pair 189F/ mb661r; forward: 5′-GGNGACCGGGATTTCTGG-3′, reverse: 5′-
CAGGMGCAACGTCYTTACC-3′; Costello and Lidstrom 1999) was performed in a CFX Connect real-time PCR system (Bio-Rad, Hercules, CA) in a final volume of 10 µl, consisting of 5 µl 2x Luna Universal qPCR Master Mix (New England
BioLabs GmbH, Frankfurt am Main, Germany), 0.7 µl forward and reverse primers each (10 µM), 0.5 µl bovine serum albumin (1%), 1.1 µl nuclease-free water and 2 µl template DNA. The thermal profile consisted of an initial denaturation step at 95°C
for 5 min, 40 cycles of denaturation at 95°C for 30 s, annealing at 62°C for 30 s, elongation at 72°C for 45 s, and an additional data acquisition step at 79°C for 8 s, followed by final elongation at 72°C for 5 min. The template DNA was used in five times
or ten times dilution and spiked with the standard to a concentration of $10^5$ copies per µl to correct for inhibition. Standards consisted of a dilution series ($10^1 – 10^6$ *pmoA* gene copies) of a PCR product flanking the *pmoA* gene of *Methylomonas*
*rhizoryzae* GJ1 (Japan Collection of Microorganisms, JCM 33990) amplified with a designed primer pair (forward: 5'-GTACGCATACGCATGAACGC-3', reverse: 5'-GTTTCCCGTGCGTTTGACTG-3'). The amplicon specificity was
confirmed using a melt curve and agarose gel electrophoresis. Samples that did not show this specificity, i.e., Forest samples, were not considered to calculate *pmoA* abundances.

# 3 Results

**3.1 Methane emission**

In total 64 out of 206 single measurement points, from both well and reference sites, showed methane emissions to the atmosphere (Figure 3, Table S2). However, only 32 fluxes were higher than 1 nmol $CH_4$ $m^{-2}$ $s^{-1}$ and 31 of these were on the Peat sites. The absolutely highest flux was 540 nmol $CH_4$ $m^{-2}$ $s^{-1}$ on the Peat site (position 16, site R-WA 264) and the highest methane uptake was –4.4 nmol $CH_4$ $m^{-2}$ $s^{-1}$ at the Forest site R-WA 273, position 14 (Table S2). Compared to the Meadow site (~14%) and Forest sites (~15%) the Peats sites (~58%) had the highest number of sample points with methane emissions flux, too (Figure 3).

The reference grid on the Peat site showed always (on three different measuring campaigns, Table S2) substantial methane emissions ranging from 15 to 380 nmol $CH_4$ $m^{-2}$ $s^{-1}$ but only at the northern and middle transect lines. The southern three points always represented a sink or the methane emissions were lower than 0.2 nmol $CH_4$ $m^{-2}$ $s^{-1}$.

As a simple first approximation, we averaged all measuring points of the individual well and reference grids (mean and median, Table 2) with all the peat sites showing net methane emissions. The Peats reference sites had the highest mean emissions (~109 nmol $m^{-2}$ $s^{-1}$). However, this should not be directly compared to more sophisticated emission techniques, e.g. long term eddy covariance studies, but rather as a snapshot of our study site for internal comparison of wells/references and different grounds (Forest, Meadow, Peat).

Mean and median values that are close to each other are typical for symmetrical distributions with minimal outliers. This holds for the data from the Forest and Meadow for both well and reference site (Table 2, Figure 4a, d). The data from the Peat sites show means that are much higher than medians indicating positively skewed data i.e. outliers on the high end (compare histogram Figure 4g). However, as such outliers can control the methane emissions of an area the mean is more suitable for an emission estimation. The difference between median and mean indicate the huge variation in methane emissions at the Peat sites, which is better visible in the Box-whisker plots (Figure 5). This is particularly evident at R-WA 264 with one grid point showing 560 nmol $CH_4$ $m^{-2}$ $s^{-1}$ and only two additional points with 30 nmol $CH_4$ $m^{-2}$ $s^{-1}$. All other 14 values are negligible small positive or representing a sink. Thus, the median of this grid is negative whereas the mean is positive (38 nmol $CH_4$ $m^{-2}$ $s^{-1}$).

As methane emissions did not show apparent differences between well and reference sites we first used the Kruskal-Wallis-Test to test for normal distribution, which the methane fluxes did not show. The Mann-Whitnes-U-test was then used to compare well and reference sites data. For R-WA 211, R-WA 209, R-WA 273, R-WA 264, R-WA 272, R-WA 275 well and reference sites were similar with regard to methane fluxes using this test. R-WA 274 and R-WA 254 showed significant differences in fluxes between well and reference sites. The reference site of R-WA 254 showed higher methane emissions than the well site. In case of R-WA 274 both sites were net methane sinks, however, the methane uptake of the well site was higher. The box-whisker plots (Figure 5, Table S9) depict this graphically. Especially, the huge differences between the Peat and the other sites is apparent.

Summarizing, all three well sampling grids, for which we observed overall methane emissions based on the mean values of 17 grid points covering an area of 900 m² around the well, were located in the Peat area. At wells R-WA 254 and R-WA 264
highly localized methane emissions with high flux rates occurred. These singular grid points with high methane emissions are not spatially correlated with the well location. Moreover, averaged methane emissions (both mean and median) were even
consistently higher at the Peat reference site than well sites in the Peat area (Table 2). All four Forest wells were a stronger sink than the corresponding reference sites at the day of measurement. The Forest sites acted as a higher methane sink than the
Meadow site.

       In addition to these sampling grids, we sampled a transect through a point with high methane emissions (Figure 6). The

resulting methane fluxes varied more than two orders of magnitude over the distance of less than one meter, whereas $CO_2$ emissions showed fewer changes and varied in total only by a factor of ~2.

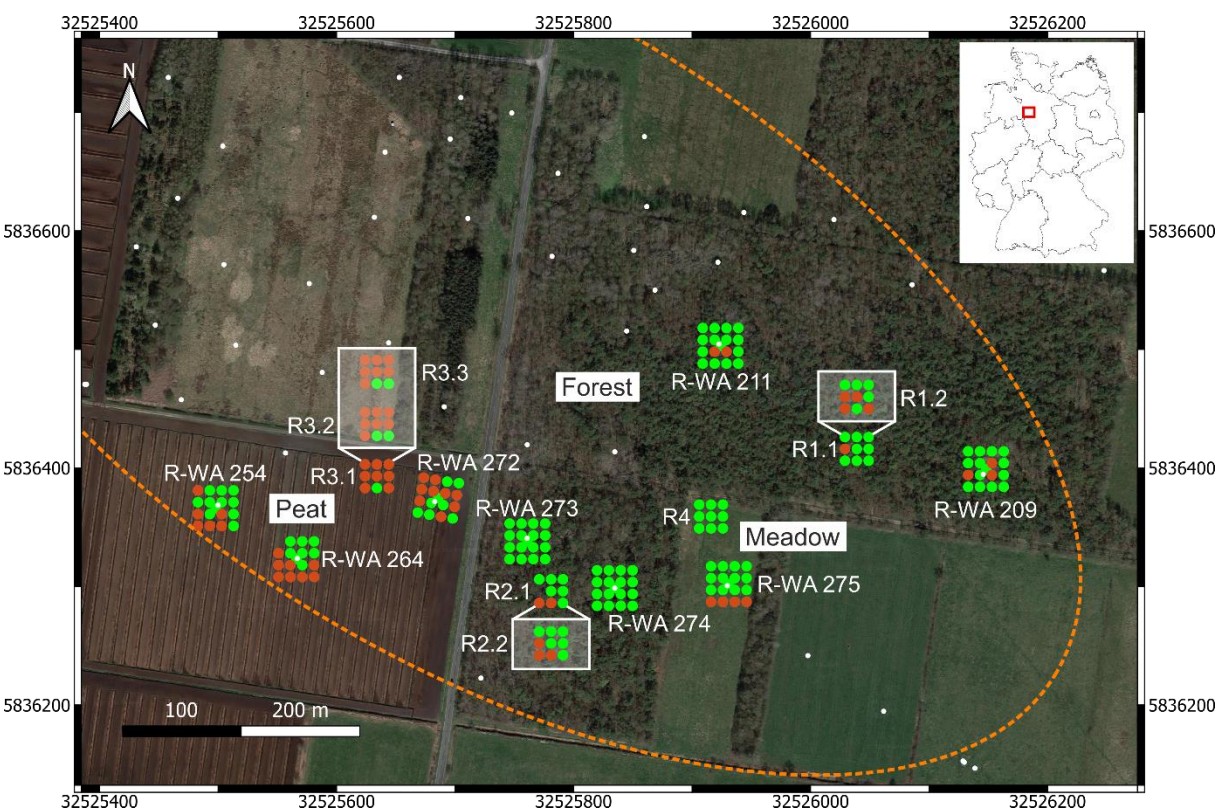

**Figure 3:** Overview of the study site indicating methane emissions (red) and methane uptake (green) for well sites (17 measuring points) and reference sites (9 measuring points), respectively. Abandoned wells are depicted in white dots. The rough dimensions of the oil field
Steimbke-Nord are outlined (orange dotted line). Coordinates are stated in UTM-32U (WGS84) with easting and northing planar coordinates in meter. The multiple measurements of reference sites are shown in a white box. The map was created using QGIS (v.3.22.3) and © Google
Earth satellite images from 2015 as background.
**Table 2:** Summary of the sampled oil well and reference sites. The displayed natural fluxes are examples from literature (Abdalla et al. 2016, Oertel et al. 2016) as well as the emissions from abandoned wells, which were compiled from Williams et al. (2021) and Cahill et al.
382   (2023).

| short name | date | area | $CH_4$ flux [nmol m$^{-2}$ s$^{-1}$] mean | median | mean soil $CH_4$ [ppm] | mean $\delta^{13}$C-$CH_4$ [‰] | mean $\delta^2$H-$CH_4$ [‰] |
|---|---|---|---|---|---|---|---|
| R-WA 211 | 09.03.2022 | Forest | –0.47 | –0.13 | 1.4 | –51 | |
| R1.1 | | Forest | –0.12 | –0.09 | 2.1 | –49.6 | |
| R-WA 209 | 10.03.2022 | Forest | –0.35 | –0.16 | 1.6 | –56.3 | |
| R1.2 | | Forest | –0.08 | –0.05 | 2.1 | –49.6 | |
| R-WA 273 | 30.03.2022 | Forest | –1.31 | –1.22 | 1.4 | –48.3 | |
| R2.1 | | Forest | –0.76 | –0.87 | 5.2 | –56.1 | |
| R-WA 274 | 31.03.2022 | Forest | –1.41 | –1.14 | 20.3 | –61 | |
| R2.2 | | Forest | –0.51 | –0.43 | 6.7 | –58 | |
| R-WA 275 | 21.04.2022 | Meadow | –0.2 | –0.2 | 3,695 | –85.4 | –222.8 |
| R4 | | Meadow | –0.1 | –0.1 | 4,467 | –99.1 | –181.8 |
| R-WA 272 | 20.04.2022 | Peat | 25.38 | 0.31 | 376,918 | –58.4 | –338 |
| R3.1 | | Peat | 50.07 | 15.42 | 181,802 | –64.9 | –306.9 |
| R-WA 254 | 27.04.2022 | Peat | 0.25 | –0.08 | 286,312 | –66.1 | –332.1 |
| R3.2 | | Peat | 109.03 | 55.79 | 369,909 | –63.1 | –316.3 |
| R-WA 264 | 28.04.2022 | Peat | 37.56 | –0.05 | 537,317 | –64 | –314.1 |
| R3.3 | | Peat | 50.5 | 20.91 | 290,555 | –65.9 | –304.1 |

natural forest flluxes      –1.9 to 23 nmol m$^{-2}$ s$^{-1-1}$
natural grassland fluxes      –0.7 to 0.8 nmol m$^{-2}$ s$^{-1}$
natural wetland fluxes      –0.5 to 650 nmol m$^{-2}$ s
abandoned well fluxes      30 to 8 x 10$^5$ nmol m$^{-2}$ s$^{-1}$

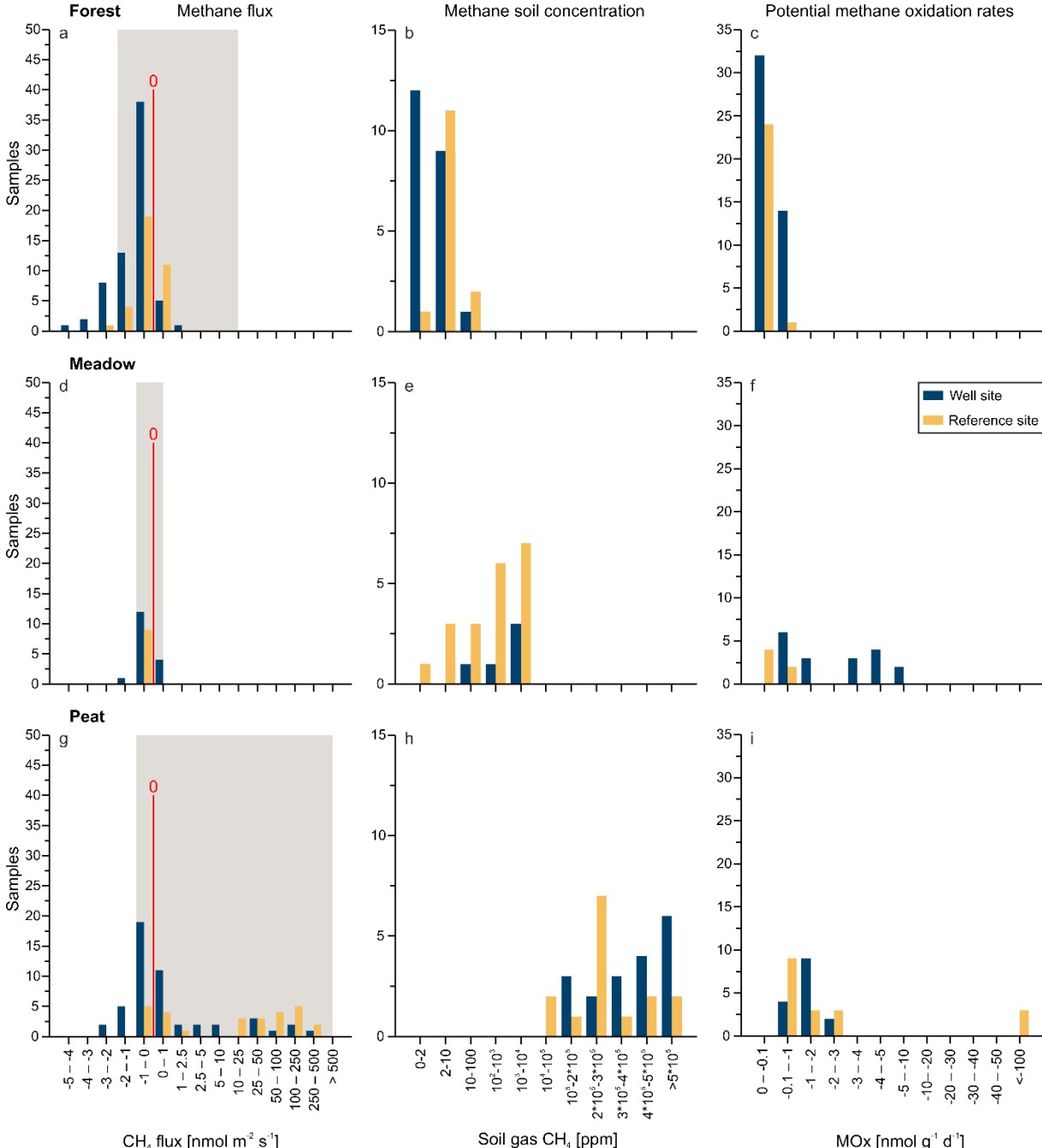

**Figure 4:** Methane flux (a, d, g), soil gas methane concentration (b, e, h), and potential methane oxidation rates (MOx; c, f, i) depicted as histograms for well (blue) and reference sites (orange) at the three areas Forest (a, b, c), Meadow (d, e, f), and Peat extraction site (g, h, i). The red line in a, d, g indicates zero flux, sites left of the line acted as net methane sinks and at the right as net methane sources. The grey background represents natural ranges mentioned in literature (Abdalla et al. 2016, Oertel et al. 2016).

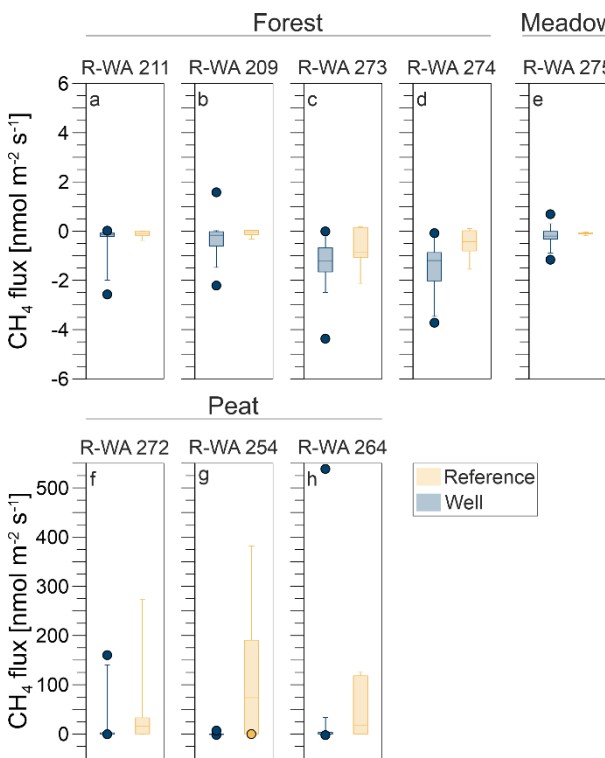


**Figure 5:** Box-whisker plots (including outliers) of methane emissions from well (blue) and reference (orange) sites: Forest (a, b, c, d), Meadow (e), Peat (f, g, h). The underlying statistical parameters are listed in Table S9.



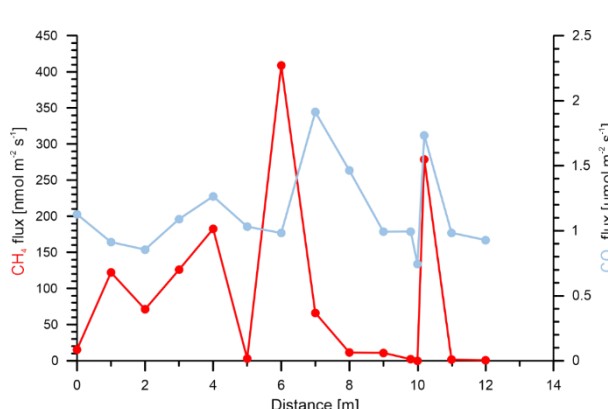

**Figure 6:** Methane (red) and $CO_2$ (blue) fluxes on a meter scale over a 12 m transect at the peat reference site. The fluxes were measured over the course of 3 h. Data is listed in Table S3.



## 3.2 Soil gas geochemistry

Soil gas samples were taken from up to 95 cm depth and analyzed in the laboratory for gas compositions including gaseous hydrocarbons ($C_1$–$C_6$) as well as carbon and hydrogen isotopic composition if concentrations were sufficient. Depths of the soil gas sampling differed and were limited by the depth of the groundwater table at the time of sampling. Generally, the sampling depth was closely above the groundwater table and is, thus, an indirect measure of the deepest interval of the vadose zone at the time of sampling. The soil methane concentrations between the sampled areas were clearly distinct, with Forest soils showing the lowest methane concentrations compared to Meadow and Peat (extraction site) soil gases (Figure 4b, e, h and 7b). The majority of methane concentrations at the Forest sites were around or below atmospheric concentrations (Table S1), however, two samples had with ~93 ppm and ~64 ppm elevated methane concentrations. These Forest sites did not emit substantial amounts of methane (Table 2). The overall mean for samples from Forest soil was ~7.5 ppm methane (Table S1), the median however was ~2.1 ppm. Soil methane concentrations in samples from the nearby Meadow site started at ~1.8 ppm and reached up to 9,200 ppm. The respective mean methane concentration was ~1,960 ppm and the median ~710 ppm. Soil gas samples from the Peat extraction site showed both, the highest overall concentration with nearly 65% methane (~645,000 ppm) and with ~315,000 ppm (mean) and 282,000 ppm (median) the highest mean and median concentration, respectively. The general differences in the soil gas composition between the three sampling areas becomes also clear from the plot of $O_2$, $CH_4$, $CO_2$, and $N_2$ concentrations with depth (Figure 7a–d).

We also analyzed the $\delta^{13}C$-$CO_2$, $\delta^{13}C$-$CH_4$, and $\delta^2H$-$CH_4$ for most soil gas samples (Table S1). Methane concentrations in the Forest soil were, however, too low to determine $\delta^2H$-$CH_4$. As for $\delta^{13}C$-$CO_2$, isotopic compositions' of Forest and Meadow soil gases were similar, ranging both between –21.7‰ and –24.9‰ (Figure 7). Soil gases from the Peat site, on the contrary, were much more $^{13}C$-enriched with $\delta^{13}C$ values up to –1.8‰ and a mean of ~–11.6‰. Thus, while for the Forest and Meadow area $\delta^{13}C$-$CO_2$ in the soil gas was relatively uniform and typical for common soil gas, variations at the Peat extraction sites were high, indicative for different controls on soil $CO_2$ in this area (Figure 7f). The $\delta^{13}C$-$CH_4$ signatures differed between all three areas with the methane in the Meadow soil being most $^{13}C$-depleted with a mean $\delta^{13}C$ value of –86.6‰, in the Forest soil of –57.4‰, and in Peat soil of –63.8‰ (Figure 7e). The mean hydrogen isotopic composition of methane differed strongly between the Meadow and Peat soil gases with $\delta^2H$-$CH_4$ of –270‰ and –320‰, respectively (Table S1). All isotope data from the reference and well sites were not systematically different from each other.

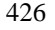

**Figure 7:** Depth profiles of $O_2$ (a), $CH_4$ (b), $CO_2$ (c), and $N_2$ (d) soil gas concentrations, as well as $\delta^{13}$C-$CH_4$ (e) and $\delta^{13}$C-$CO_2$ (f) values for
Forest (brown diamonds), Meadow (light green squares) and Peat (dark grey circles) sites. Atmospheric values are depicted as blue lines.
Note the logarithmic scales in a to c. Isotopic composition of methane (e) and carbon dioxide (f) is depicted relative to the Vienna Pee Dee
Belemnite (VPDB) standard.


### 3.3 Methane oxidation rates

Methane oxidation rates were determined to investigate the soils' potential to mitigate methane emissions. In total 27 positions
were sampled in up to two depths, resulting in 46 methane oxidation rates. Mean methane oxidation rates per g dry soil (Table
3) were lowest for Forest soils (~ 0.04 nmol $g^{-1}$ $s^{-1}$) and highest for soils from the Peat sites (~18.3 nmol $g^{-1}$ $s^{-1}$) with
intermediate values for Meadow soils. To get an estimate of actual oxidation rates in the soil column, we calculated the
potential methane oxidation rates per g wet soil sample and a height of 20 cm (Table 3, Table S4). These rates followed the
same pattern as the dry methane oxidation rates and methane soil concentrations and were highest in the industrial Peat area.

**Table 3:** Mean areal methane oxidation rates (MOx) for Forest, Meadow and Peat sites calculated per gram dry soil as well as for dry and wet soil of a volume of 1 m$^2$ and 0.2 meter height (= 0.2 m$^3$) as well as mean 16S-RNA gene and *pmoA* abundance per gram dry soil. *pmoA* abundance was calculated as relative to 16S rRNA gene abundances.

| | MOx dry [nmol CH$_4$ g$^{-1}$ s$^{-1}$] | MOx dry [nmol CH$_4$ 0.2 m$^{-3}$ s$^{-1}$] | MOx wet [nmol CH$_4$ 0.2 m$^{-3}$ s$^{-1}$] | 16S rRNA gene [10$^9$ g$^{-1}$ dry wt.] | *pmoA* [10$^6$ g$^{-1}$ dry wt.] | *pmoA* abundance [%] |
|---|---|---|---|---|---|---|
| Forest | 0.04 | 85 | 47 | 13 | | |
| Meadow | 1.4 | 2475 | 3106 | 16 | 30 | 0.19 |
| Peat | 18.3 | 18199 | 14114 | 4.6 | 14 | 0.30 |

For a selected experiment on the methane turnover in the Peat area the carbon isotopic fractionation of methane during aerobic methane oxidation was determined in the laboratory (see supplement S3). Using a calculation from (Feisthauer et al., 2011) this resulted in an epsilon (ε) of –31.3‰ (Supplement S3).

### 3.4 MOB abundance and identification

We determined MOB abundances by targeting both, the general 16S rRNA gene and the for the methanotrophic bacteria specific *pmoA* gene using qPCR (Table 3). The Peat sites had with ~4.6 x 10$^9$ copies g$^{-1}$ dry weight about three times lower 16S RNA gene copies than the other two sites with 1.3 x 10$^{10}$ (Forest) and 1.6 x 10$^{10}$ (Meadow) copies g$^{-1}$ dry weight. The *pmoA* gene abundances were similar at Meadow and Peat site, with 3.0 x 10$^7$ and 1.4 x 10$^7$ copies g$^{-1}$ dry weight, respectively. The relative abundance of the *pmoA* gene was highest in the Peat (~0.30%) reaching up to 0.89%, followed by the Meadow (0.19%). However, there were huge differences between the samples in each area (Table S5).

We used DNA-based microbial analyses to identify changes in bacterial community over depth and identify potential methanotrophic key players. Bacterial 16S rRNA gene sequencing revealed between ~1.5 x 10$^4$ and ~1.35 x 10$^5$ sequences per sample with a median of ~8.5 x 10$^4$ sequences and a mean library coverage C of >98.5% (data not shown). In total ~22 x 10$^4$ ZOTUs were determined. A comparison on genus level with published taxa known to contain the *pmo* operon sequences resulted in up to 151 potential methanotrophic ZOTUs, grouping into 15 methanotrophic genera and 5 families (Table S6). The most abundant putative methanotrophic family in amplicon libraries was *Methylacidiphilaceae*, with 71 uncultured ZOTUs followed by *Beijerinckiaceae*. The most abundant genera were *Methylocystis* and the uncultured cluster SH765B-TzT-35 from the *Methylomirabilaceae* family (hereafter referred to as SH765B-TzT-35). In the following, we grouped the ZOTUs belonging to the same genera together in order to simplify the dataset and make changes between the areas better visible.

Most reads affiliating with known methanotrophic taxa reads were found at the Peat site, whereas Forest and Meadow had about half as much reads. In Forest samples, most of such reads were found in the top layer. On the contrary, they increased with depth for the Meadow site until a depth of 8–13 cm and 15–20 cm at the Peat site and decreased afterwards in both cases slightly (Table S6). The top layer at Forest and Meadow sites was with regard to methanotrophic taxa dominated by an uncultured *Methylacidiphilaceae* genus, which relative contribution to all reads decreased with depth (Figure 8). A member of

the genus *Methylocystis*, however, dominated the peat site. Its relative abundance first increased to a depth of 20 cm and then abruptly declined at a depth of more than 40 cm. In samples of 40 cm and below SH765B-TzT-35 dominated the methanotrophic community (Figure 8).

In addition to bacterial 16S RNA gene sequencing, we used archaeal primers to identify methanogenic key players. Sequencing resulted in ~9.3 x $10^3$ and ~ 1.2 x $10^5$ reads per sample with a coverage of >99.9% (data not shown). Overall, 798 ZOTU were identified and a comparison with known methanogenic genera revealed 132 potential methanogenic ZOTU (Table S7). These could be grouped into 11 genera and 9 families (Figure 8). The most abundant genera were *Methanosarcina*, followed by *Methanoregula*, which was almost exclusive present in Peat samples, and third *Methanosaeta*. Together with *Methanobacterium* they account for 96% of methanogenic reads over all samples.

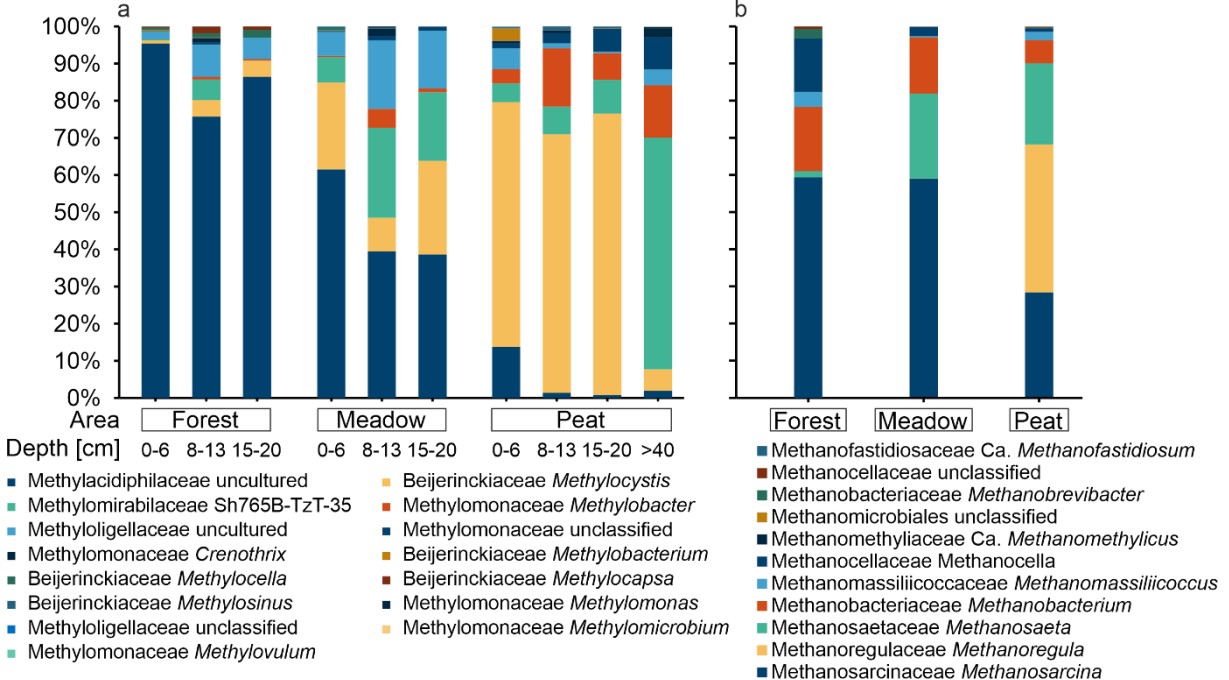

**Figure 8:** a) Relative abundance of potential methanotrophic genera estimated at three depth intervals detected in the Forest, Meadow, and at four depth intervals in the Peat areas. b) Potential methanogenic genera detected at the three areas Forest, Meadow, and Peat depicted as relative abundance. Reads are displayed as relation to the sum of all reads associated with methanotrophic (a) or methanogenic (b) taxa in the respective sample pool (well and reference sites together).

## 4 Discussion

### 4.1 Sources of methane in soil and emitted gases

We evaluated methane emission in the described complex and organic rich setting by combining the methane fluxes and soil gas geochemistry. Although these emissions were detected at both the well and the reference sites, it was unclear whether they originated from a leaking well or from methanogenesis. Thus, we determined the isotopic composition of methane ($\delta^{13}$C-CH$_4$ and $\delta^2$H-CH$_4$) to distinguish between a thermogenic (in our case oil-associated) and a biogenic source of the methane emissions (Schoell 1980, Whiticar, 1999; Milkov and Etiope, 2018). Thermogenic gases, which are produced during the maturation of organic material and which occur in natural gases and oil-associated, are characterized by relatively high $\delta^{13}$C values (> ~–50‰). In combination with $\delta^2$H values of the methane, thermogenic origins can be well recognized in $\delta^{13}$C/$\delta^2$H diagrams (Figure 9a). Furthermore, we included measurements of the same parameters at reference sites to determine the natural methane-related biogeochemical background. This approach (see below) helps to get information on whether well-integrity issues, migration of biogenic methane along the well may have taken place (e.g., Vielstädte et al., 2015; 2017) or natural biogenic methane sources and processes in the upper soil are responsible for the methane fluxes.

Using our emission measurements, we could identify three well sites and their respective reference measurements with net methane emissions (Figure 5, Table 2), all of which were located at the Peat site. The first indication that the methane emissions were not oil well-related was that the single peat reference site (measured at three different days) emitted more methane than the corresponding well sites (Table 2). Further, all peat soil gases contained >5% methane with a median of ~35% with no recognizable trend between sites (Table S1). However, none of the methane samples from Steimbke showed an isotopic signature typical for thermogenic methane. Together, this excludes the leakage of relevant amounts of natural gases from the oil reservoir to the atmosphere or upper soils in Steimbke. Finally and supporting this conclusion, oil-associated gases and natural gas contain substantial amounts of ethane and other higher hydrocarbons, which were only found in trace amounts in the analyzed gases (Table S1). Both gases (ethane and propane) can be produced in such trace amounts as byproduct during methanogenesis and are typically associated with high amounts of biogenic methane (Schloemer et al. 2018, Oremland et al. 1988). Methane concentrations were not sufficient for $\delta^2$H analyses in all gas samples, so that the following does not necessarily hold for low concentrated samples. However, our $\delta^{13}$C/$\delta^2$H data indicate that in Steimbke the biogenic methane was formed through methanogenesis using acetate (methyl fermentation; acetoclastic) or $CO_2$-reduction (Figure 9b).

Another previously proposed test for well leakage focused on soil gas composition (Romanak et al., 2017; Romanak et al., 2014). The authors argue that the oxygen and carbon dioxide concentrations in soil gases, driven by normal microbial respiration, should sum-up to around 21%. An excess in $CO_2$ would hint towards an additional $CO_2$ source (Romanak et al., 2012). Therefore, they suggest methane from a leaking well, which is oxidized to $CO_2$, could be such a source or in their investigated case directly leaking $CO_2$ from a CCS storage site. We observed such enhanced $CO_2$ concentrations in the Peats soil gases (Figure 9c). Forest measurements and the majority of the Meadow followed, in contrast either a conversion of 1:1

(respiration) or 2:1 oxygen to $CO_2$, with the latter corresponding to the stoichiometry of aerobic methane oxidation (Romanak et al., 2012; and references therein). The Peat soil gas compositions spread between both processes and conversions. About
half of the samples, however, were enriched in $CO_2$ (up to 33%). In our view, the drastically increased $CO_2$ levels in Peat soil gases could be best explained by an extensive degradation of peat by hydrolysis and fermentation to acetate and fatty acids.
Those compounds are hereby subsequently converted to methane and $CO_2$ by acetoclastic methanogenesis with possible contributions of methanogenic conversion of $H_2$ and $CO_2$ to methane (e.g. Conrad, 2012). This is supported by the Peat's high
methane and $CO_2$ concentrations (Figure 9c). This methane is than oxidized by MOB to $CO_2$, which further complicates the soil gas interpretation.
While our approach cannot exclude well integrity problems in general, our data argue against methane leakage into the upper soil and/or atmosphere from the reservoir for the studied eight wells in the Steimbke-Nord oil field. Furthermore, the high
methane emissions at both, well and reference sites, argue against the migration of shallow biogenic methane along the wells (methane concentrations were not higher in the well grid than in the reference grid samples). A comparison with other reported
fluxes underlines that the here determined fluxes (Figure 4) are in the range of natural methane emissions ($-2 – 600$ nmol m$^{-2}$ s$^{-1}$, Abdalla et al. 2016) and the lower end of emission rates from abandoned wells ($\sim30$ nmol s$^{-1}$ – 800 µmol s$^{-1}$, Cahill et al.
2023, Williams et al. 2021). Overall, we join previous studies in a call for better surveillance of abandoned wells past abandonment (Cahill et al. 2023, Riddick et al. 2020), more standardization and a comprehensive approach for assessing
fugitive gas migration in the field (Samano et al. 2022), and propose the here introduced approach. Furthermore, we used this data to look into the apparent differences in methane cycling between the three sites, which will be discussed in the following.

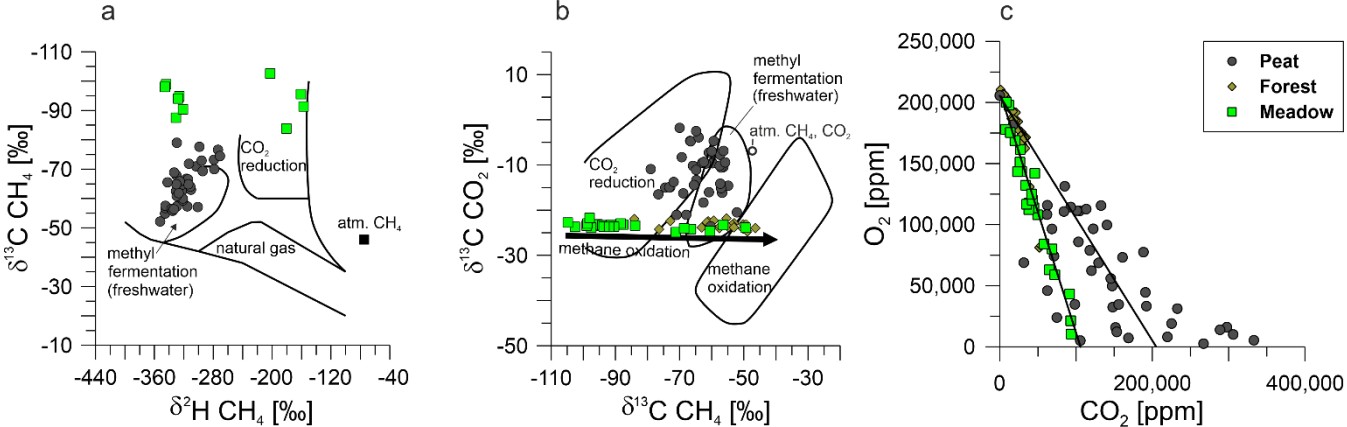

**Figure 9:** Cross-plots of soil gases, namely isotopic composition of a) methane with regard to stable isotopes of carbon and hydrogen as well as b) methane and carbon dioxide to characterize the methane's sources, c) oxygen and carbon dioxide, for the three sites Peat (dark
grey circles), Forest (brown diamonds), and Meadow (light green squares). The lines represents in c) left, the consumptions of oxygen via methane oxidation and right for normal soil respiration. Isotopic composition a) and b) after Whiticar (1999). Comparison of oxygen and
$CO_2$ in panels c) after Romanak et al. (2012).

## 4.2 Natural methane-cycling at the study sites

In the three peat rich surveyed vegetation types in Steimbke, differences in soil gas methane concentrations were more pronounced than in methane emissions. These emissions, however, showed high spatial variations and tended to change from source to sink between two measuring points and, thus, on short distances and eventually over time. We therefore conducted a second sampling campaign at the Peat extraction reference site with flux measurements only one meter or less apart to better understand variations on a smaller scale than the one usually chosen in our study (10 x 10 m). With this new approach, we observed high spatial heterogeneity of the methane emissions (Figure 6), which are in agreement with other soil studies (Davidson et al., 2002; Savage et al., 2014; Ambus and Christensen, 1995; Le Mer and Roger, 2001).

Distinct controls for both, spatial and longer temporal variability could not be resolved in our study but could be explained by changes in soil compaction (Flechard et al., 2007), differences in moisture content (Basiliko et al., 2007), fluctuating macropores (Schwen et al., 2015), differing floras (Jentzsch et al., 2024), microforms (Welpelo et al., 2024) and fauna (Lubbers et al., 2013). Furthermore, precipitation and air pressure variations (i.e., barometric pumping, Forde et al. 2019b) between two consecutive measurements could have affected emission patterns and rates as well (Blagodatsky and Smith, 2012). To address short-term temporal variation, we use the measurements of reference sites in the Peat, which have been visited three times (Figure 3). First on April 20, 2022, and then one week later on two consecutive days (April 27 and 28, 2022). Thereby the overall flux-pattern at the reference site changed from one week to another (Figure S5a, b, c) and fluxes at the same spot differed in part greatly. The fluxes at one point on two consecutive days differed less (Table S2) and the overall pattern remained similar. In contrast, soil methane concentration did not vary as much over time as the respective methane fluxes (Table S1). Compared to methane, $CO_2$ fluxes at the same spots were much more stable and did not show a time dependent variation (Table S2). This temporal data and the whole data set underline the importance of individual reference measurements and that single measurement points are not sufficient to evaluate background emissions properly.

In natural environments, biogenic methane emissions are the result of the net balance between production and consumption, and the biotic regulation of emissions can occur at the methanogenic and methanotrophic side. Regarding methane production, previous studies discussed the following possible factors to control methanogenesis in peatlands (1) availability of acetate due to acetate-producing bacteria outcompeting $CO_2$-reducing methanogens (Kotsyurbenko, 2005), (2) phenolic compound concentrations, which might limit peat degradation (Freeman et al., 2001), (3) and temperature (Brauer et al., 2006). We assume that one or more of these controls are also responsible for the presumably different predominating methanogenic pathways indicated by the molecular community analyses and the isotopic compositions of methane in our studied Peat and Meadow areas. At all sites both acetoclastic and hydrogenotrophic methanogens were present (Figure 8). The genetic analyses suggest a higher methanogenic potential at the Peat site, as there were relatively more methanogenic reads found and a higher diversity of methanogens (Figure 8, Table S7). The acetoclastic methanogenic generas were mostly *Methanosarcina* and *Methanosaeta* whereas the also observed *Methanoregula* and *Methanobacterium* are hydrogenotrophic methanogens (Conrad, 2020). Soil temperatures were similar at the point of sampling with ~10°C, which supported the growth of both acetoclastic

and hydrogenotrophic methanogens. Our isotopic data shown in Fig. 9a, b underline differences between the sites and suggest that methane was produced via different methanogenic pathways. The methane at the Peat sites seems to be mostly derived from acetate in contrast to $CO_2$ reduction as the main methanogenic pathway at the Meadow site. This is underlined by the higher mean $\delta^{13}C$-$CO_2$ in Peat soil gases (–12‰) compared to soil gases from Meadow and Forest sites (–23.5‰). The first indicate that substantial amounts of $CO_2$ in Peat soil gases have resulted from fractionating acetoclastic methanogenesis increasing the pool of relatively $^{13}C$-enriched $CO_2$ (Corbett et al. 2012). Methane concentrations in the forest soils were insufficient for $\delta^2H$-$CH_4$ measurements. However, the $\delta^{13}C$-$CO_2$ and $\delta^{13}C$-$CH_4$ data suggest that $CO_2$ reduction represents the primary methanogenic pathway, with a pronounced isotopic alteration observed in samples with low methane concentrations due to methane oxidation. One explanation for the site dependent difference of the predominant methanogenic pathway could be the differences in peat degradation progression due to the removal of vegetation for peat extraction. The drainage of peatlands is known to lead to decomposition of peat and results in substantial losses of carbon (Couwenberg, 2011). This may in part also explain the higher methane emissions from the active peat extraction site (Peat site) as the drainage of the investigate area started already decades ago. However, the peat extraction itself started at this site only recently (2017/18). Thus, we expect that the decomposition of deeper peat layers and the remaining peat intensified after the start of the extraction. Furthermore, about 1 m of peat was already mined from the whole area used for extraction. This extraction, led to a lowering of the terrain surface (compared to the surroundings) and consequently to a relatively higher water table, which is one of the main factors for higher methane emissions (Abdalla et al., 2016) as it limits the penetration of oxygen into deeper layers necessary for methane-oxidizing bacteria (Basiliko et al., 2007). For our gas geochemical study and related sampling strategy, however, a respective in-depth understanding of the drivers for the individual methane-formation pathways was beyond the scope of our study and thus we will in the following focus on the microbial methane filter.

The observed relatively low potential methane oxidation rates at the Forest sites could result from high affinity methanotrophs, which are specialized to low methane concentrations in well-aerated soils (Bengtson et al., 2009; Kolb, 2009). The very high rates at the Peat site in contrast are an indication of low affinity methanotrophs, which require higher methane concentrations (>100 ppm, Whiticar 2020, and references therein). In combination with these differences in potential methane oxidation rates, our phylogenic data suggests that members of the *Methylacidiphilaceae* family correspond to high affinity methane oxidation, whereas *Methylocystis*, *Methylobacter*, and Sh765B-TzT-35 predominate at higher methane concentrations. Kaupper et al. (2021) compared pristine and restored peatlands previously observed a similar shift from *Methylacidiphilaceae* to *Methylocystis* between both settings. So the microbial community at our studied Forest sites (with peat underneath), which consisted mainly of *Methylacidiphilaceae*, was more similar to that of a pristine peatland than the communities at the other vegetation type settings. The community of the active peat extraction site, which was dominated by *Methylocystis* showed, in contrast, higher similarity to the restored site in Kaupper et al. (2021). This indicates that starting with peat drainage the composition of the methanotrophic community changes but remains active throughout the peat extraction process. Our phylogeny analysis were supported by phospholipid fatty acid (PLFA) analyses of selected samples from the Peat site (Supplements S2), which indicate that the species *Methylocystis heyeri,* a Type II (α-Proteobacteria), was likely involved in

methane oxidation at these sites (Figure S2). We also found these PFLA in incubation samples and observed a significant increase after methane addition.

Especially interesting is that we detected sequences of the genus Sh765B-TzT-35 in deeper and probably anoxic peat layers, which belongs to the family *Methylomirabilaceae*. Other members of this family are known to oxidize methane under anaerobic conditions by internal oxygen production from nitrite reduction to dinitrogen (Ettwig et al., 2010; Versantvoort et al., 2018). Although this internal oxygen production was so far not demonstrated for species of Sh765B-TzT-35, a previous study showed their ability to anaerobically oxidize methane (Nakamura, 2019), which hints towards the same or at least similar mechanisms also for this genus in the studied Peat site.

The discussed methanotrophic community resulted in the highest methane oxidation rates in samples with elevated soil methane concentrations (>4000 ppm), which is in concordance with previous studies (Basiliko et al., 2007; Moore and Dalva, 1997). It was recently shown, that in addition to substrate availability (here methane concentration), the methanotrophic community can be influenced by physico-chemical parameters and land use (Kaupper et al., 2022 and references therein). Kaupper et al. (2022) showed that the environmental parameters, total C and N content, and electrical conductivity, indicative of salinity, affected the active bacterial community. This suggests that the methanotrophic communities can adapt to different methane regimes and, as speculation, could mitigate an occurring potential methane leakage from an underlying abandoned well over time. Converting our values for mean methane emissions to enable the comparison with literature data, we observed an emission rate of ~23 g m$^{-2}$ yr$^{-1}$ for the Peat sites. These numbers are in our case without emissions from ditches, which Sundh et al. (2000) showed can be substantial. An in-depth study on the influence of vegetation on methane emission conducted by Welpelo et al. (2024) at a rewetted peat site about 3 km north-west from our study area, estimated yearly emission between 7.1 and 36.1 g m$^{-2}$ year$^{-1}$. As our field campaign was conducted in April 2022, and we observed comparable methane emissions to their combination of measurement and modeling for the same season, our estimation seems plausible, although, the Peat's groundwater table was comparably lower. The emissions at the Peat extraction site's (our study) were about twice (Strack et al., 2016) to more than hundredfold (Wilson et al., 2016) higher than from pristine peat sites and about tenfold higher than from a restored peatland (Strack et al., 2014). The carbon dioxide emissions (Table S2) from the Peat sites were similar to another unrestored peat extraction site (Strack et al. 2014). In addition, it is possible that the progressed peat extraction provided a different type and quality of organic precursor substrates than the Forest and Meadow sites as suggested from and observed in other peat sites (Alstad and Whiticar, 2011). Our data suggest that active peat extraction sites can be significant methane sources and that these areas do not necessarily emit less methane than rewetted ones as stated in literature (Welpelo et al., 2024; Bieniada and Strack, 2021; Rankin et al., 2018; Abdalla et al., 2016).

## 4.3 Extent of natural microbial mitigation of potential subsurface leakage

Our data, which is in line with another study on buried abandoned wells in the Netherlands (Schout et al. 2019), suggests that relying only on methane emission measurements to detect well leakage can be associated with the risk of missing integrity-

compromised wells. In our case, at some measuring points soil methane concentrations reached ~45% of biogenic methane at 20 cm depth, e.g., site R-WA 264, position 2, but the soils still acted as a methane sink at the surface ($-1.2$ nmol m$^{-2}$ s$^{-1}$, Table
S1, S2). This is probably due to the high methane oxidation potential in these soils due to the presence of a large population of methanotrophs (Figure 8, Table S6), which has also be reported previously (Kolb and Horn, 2012; Ho et al., 2019; Guerrero-
Cruz et al., 2021).

It remains unclear to what extent natural microbial oxidation capacities for methane could degrade upward migrating methane

in the soil in the event of a broken gas or oil well. This will most likely be less efficient at the beginning of a leakage, but might rapidly increase due to the adaptation of the respective microbial communities in the affected soil layers. However, such
processes could be highly relevant for Germany, as 15% of abandoned wells in Germany are located in areas with highly organic-rich soils such as peat (mostly in Northern Germany). These areas most likely already contain a microbial community
preadapted due to the naturally elevated methane concentrations. In a recent study, Schout et al. (2019) observed such a situation, as they were unable to detect any methane emissions into the atmosphere above a leaking borehole, but could show
high methane fluxes after removing the top 2 m of the soil and, thus, the microbial methane filter. This is also in-line with a study by Cahill et al. (2023), who found that 5 out of 10 surveyed wells (9 unconventional) in Canada were leaking fugitive
methane, however only two showed direct methane emission (up to 3 x 10$^3$ nmol m$^{-2}$ s$^{-1}$) whereas the other were emitting elevated levels of $CO_2$ (up to 15 µmol m$^{-2}$ s$^{-1}$). These $CO_2$ fluxes were interpreted to result from enhanced bacterial methane
oxidation mitigating the fugitive methane release from leaking abandoned wells by natural soils (here with a lower organic carbon content) and lowered the total greenhouse gas emission substantially. However, in addition to indirect indications such
as $\delta^{13}$C-$CO_2$ values (Cahill et al. 2023), our study demonstrated that it is advantageous to also determine methane oxidation rates. Here, we measured high methane oxidation capacities of (wet) peat samples in our lab of up to ~14,000 nmol CH$_4$ 0.2
m$^{-3}$ s$^{-1}$ (= 0.8 g 0.2 m$^{-3}$ h$^{-1}$, Table 3). To put this in perspective, methane leakage rates from plugged wells in two regions in Canada ranged between 0.04 to 1 g CH$_4$ well$^{-1}$ h$^{-1}$ (Bowman et al., 2023) and up to ~0.2 g well$^{-1}$ h$^{-1}$ from unconventional
plugged wells (Cahill et al. 2023) are similar in ranges. Further research is required to examine the activity and precise functioning of this microbial filter, with particular attention paid to the influence of seasonality (i.e., temperature).


# 5 Conclusion

While it is well known that abandoned oil and gas wells can have integrity issues, respective knowledge particularly on the 20,000 cut and buried wells in Germany, is lacking. Here we provide with our multi-methodological approach first data on
potential methane fluxes from abandoned oil wells to the atmosphere. We combined emission data (positive and negative) at wells and reference areas with gas geochemical characterization of soil gas samples to investigate eight wells in a peat rich
setting with three different land use types (Forest, Meadow, Peat extraction).

The Peat extraction site was the only one, which emitted substantial amounts of methane. However, in general no difference

in surface methane emission rates between well and reference sites independent of site characteristics (active peat mining, drained peat vegetated with birch trees or grassland) were observed. With respect to soil gases, the three areas showed highly
variable but spatially correlating (i.e. area specific) methane concentrations concurring with $CO_2$ concentrations. The in-depth gas and isotope geochemical analysis revealed biogenic methane as source for the net emissions at the open peat site (methyl
fermentation) and the meadow ($CO_2$ reduction pathway with partial methane oxidation). These findings and the absence of higher hydrocarbons excludes thermogenic gas emissions from the plugged wells. Overall, we conclude that there is no
connection between the methane emissions detected and the abandoned wells investigated. Furthermore, the factors discussed above suggest that the level of disturbance can be considered as the major driving force for the here shown methane emissions.
Thus, the anthropogenic influences play a key-role for methane formation and emission in such altered ecosystems.

Furthermore, the laboratory methane oxidation rates derived from our incubated peat samples demonstrated the capacity to

counterbalance reported leakage rates for buried abandoned wells in other regions. The activity of such a microbial methane filter poses the risk for false negative leakage classification. Overall the observed methanotrophy could be highly relevant for
Germany as 15% of our cut and buried wells are located in areas with very organic-rich soils. However, for a comprehensive evaluation of the situation of abandoned wells in Germany further investigations are needed. Therefore, additional sampling
at different sites (oil/gas wells of different age and deconstruction histories) in Northern Germany with the here introduced methodology are under way and we will evaluate the natural mitigation potential for different soil types and land uses.
In conclusion, exclusively using emission-based approaches are not suited for integrity failure assessments of buried wells as these would be susceptible to misinterpretations. We highly recommend a holistic approach for surveillance including the
determination of methane emission, soil gas composition and isotopic signatures at and in the vicinity of well sites against the background of a carefully selected reference site.

## 6 Acknowledgement

We thank Daniela Zoch, Daniela Graskamp, Thilo Falkenberg, Laurin Rösler, Lukas Heine, Nicole Becker, Alana Zimmer, Georg Scheeder, Dietmar Laszinski, and Christian Seeger for their help in the field and laboratory. Furthermore, we thank Christian Ostertag-Henning for fruitful scientific discussions and the local peat extraction company for repeated access to the study site. The BGR internal funding through project A-0202019.A and DFG grant HO 6234/1-2 made this study possible.

## 7 Data availability

Measured and derived data supporting the findings of this study are available in the supplementary data sheet.

## 8 Author contributions

M.B., S.S., S.F.A.J., and M.K. conceived and designed the experiments. S.F.A.J., S.S., M.B., M.K. conducted the fieldwork and performed the experiments. S.F.A.J. and T.H. performed qPCR and processed the data. Main data interpretation was performed by S.F.A.J. in cooperation with the co-authors. S.F.A.J. wrote the main manuscript text with input from M.B., S.S., M.K., T.H., and M.A.H. All authors read and approved the final version of the manuscript.

## 9 Competing interests

The authors declare no competing interests.

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
