# Peer review of "Interferences caused by the biogeochemical methane cycle in peats during the assessment of abandoned oil wells"

_EGUsphere, 2024_

## Referee Comment (RC2)

[referee-annotated manuscript omitted]

---

## Author Comment (AC1)

**Response to comments by reviewer Aaron Cahill**

This study describes an investigation that sought to evaluate the integrity of legacy oil and gas wells with a view to identifying leakage through fluxes of CH4 from soils into the atmosphere. The authors have used a powerful combined approach utilizing soil surface flux measurements, depth discrete soil gas concentration measurements with stable carbon isotope analyses and evaluated the soil microbiome to better understand the sources and cycling of CH4 at their sites which were dominated by peat/organic rich sediments. The authors describe how due to the presence of peat at their sites that a strong underlying methane cycling system was active that acted to make it very difficult to easily identify anomalous CH4 and potentially mask any leakage from the wells. That being said all data seem to indicate that the observed high levels of CH4 observed are almost certainly related to the peat rich settings and not from well leakage. This is important as >15% of the >20k onshore legacy wells in Germany are located in such peat rich settings. Consequently, being able to evaluate legacy well integrity against a backdrop of active methane cycling is of great value. It seems as though the study, which perhaps set out to evaluate well integrity and leakage, has pivoted to having to unravel the peat system and its methane cycling. Thus it seems to focus more on those aspects (as a necessity) rather than the wells and their integrity themselves. and in doing so is a very valuable case study for others to learn from where legacy wells are hosted in similar highly active methane cycling environments

Overall, the study has been well conducted with excellent fieldwork and analytical efforts. I commend the authors particularly for the nice addition and use of microbial data in trying to unravel the puzzle which is often missing in such studies.

> *We thank the reviewer Aaron Cahill for his positive view of our manuscript and the topic we are working on.*

Overall, the article is of good quality but some limitations in the written language which could do with improvement at multiple points throughout. The figures are of good quality, but a few tweaks could make them even better (See comments below).

> *We thank the reviewer again and will address the concerns mentioned by the reviewer as well as improve the written language (see below).*

I recommend publication after moderate revisions and addressing the following comments.

Comments to address as follows:

L36 -39: The discussion on what abandonment means is a little inaccurate and could be improved. In my understanding there are various definitions in different jurisdictions, but abandonment itself is a part of the decommissioning process where the well itself is first sealed/flow zones are isolated with plugs. In fact, the abandonment process (which is part of the decommissioning process) can then have several phases such as in the UK where there are 5 states of abandonment that are enacted depending on the well and what is being done with it.

See https://www.nstauthority.co.uk/media/1100/operator_work_instructions_v3.pdf . Wells can also be suspended which is a different status. Next there are other parts of the overall decommissioning process where the well head is removed, cut capped and buried (such as the wells you examine here) and in Canada for example the well is then subject to a reclamation certificate. So, you need to be more specific on the generalized process of decommissioning in a generalized sense recognizing its phases and different potential statuses. Even though the details can vary by jurisdiction, more if not all western regions follow a similar process now. Then after you do this generally it perhaps makes sense to use your study area (i.e. Germany) and describe in detail the process and definitions. In my experience no jurisdictions these days allow a well to have the head remain and call that decommissioned – it would be termed suspended or similar. I think all jurisdictions these days aim towards full decommissioning and reclamation of the land... I think your description needs improvement in general.

*We thank the reviewer for the comment and will rewrite the passage without getting too much into the technical details as this is a methodological manuscript focusing on the soil's processes.*

*Now reads: „Abandonment procedures today depend on national regulations and are now often similar although they have differed strongly in the past. However, the greatest impact on the country's abandoned well situation has probably been the extent to which these regulations were properly enforced. Some countries are additionally struggling with the situation of undocumented or orphaned wells (Boutot et al. 2022). This resulted in the current situation that in some cases only the well head was closed everything was left in place (Pekney et al. 2020, Williams et al. 2020), in others an open hole was left in the ground (Pekney et al. 2020, Lebel et al. 2020), or the wells were properly filled, cut and the remains buried (Schout et al. 2019, Davies et al. 2014, ). In Germany for example, first regulations date back to 1904 and were refined every few decades to the last update of 2006 (von Georne et al. 2010). Well integrity failures are, however, a known issue. This resulted in varying situations around the world and a general call to action as anthropogenic methane emissions need to be cut (Saunois et al., 2020). We use the term "abandoned well" here to refer to a former oil or gas well in Germany that has been decommissioned and buried in accordance with the guidelines in force at the time (von Georne et al. 2010)."*

L32: Again, I think this is a little inaccurate as other countries (most western ones) do require full reclamation including cut and buried status. If you expand more accurately as instructed above it will improve the accuracy of the manuscript.

*We agree with the reviewer and will change the passage as described above.*

L44-48: The problem you are identifying is not just for methane rich environments, but it is a result of trying to detect leakage in a dynamic and active zone of the soil system. In shallow soils, many physical and bio-geochemical factors can act to alter/attenuate any signal being measured or sought after on timescales of hours/days, meaning it might lead to false positives, positive false (and/or just make it difficult to see what is being looked for). I think you should expand this a little to explain more clearly upfront. A subset or very specific issue with this involves systems that are methane rich which are more prone to generating false positives and that is something you are particularly examining here as you have peat rich soils. I feel your introduction needs some more general discussion around the factors that influence observations associated with soil gas and efflux in shallow soil systems and oil and gas well integrity...

*We will rework the respective part of the introductions as the reviewer recommended. It now reads:*

*"For cut and buried wells (e.g., in Germany, the Netherlands, and UK), single measurements atop the wells location are insufficient (Schout et al. 2019). In this case, upwards migrating natural gas can be subject to several physical and biogeochemical processes, e.g., microbial oxidation is able to alter concentrations and even isotopic composition (Whiticar 2020). Leaking gas can migrate away from the wells location (Dennis et al 2022, Forde et al. 2022), disperse through the soil and potentially be oxidized by methanotrophic microorganisms on its way to the atmosphere (Forde et al. 2022). Thus, false negative results would be obtained. In addition, biogenic methane can be microbially produced in shallow anoxic soils by methanogenesis. Thereby, organic carbon degradation facilitated via a complex network of trophically linked microorganisms (e.g., intermediary ecosystem metabolism, Drake et al. 2009) ultimately resulting in methane production when alternative electron acceptors except for carbon dioxide are depleted (Whiticar 2020). This methanogenesis is mainly carried out by three types of anaerobic archaea in more than 30 genera: 1) acetoclastic methanogens converting acetate to methane and carbon dioxide, 2) hydrogenotrophic methanogens, reducing carbon dioxide to methane with hydrogen, and 3) methylotrophic methanogens disproportionating methyl groups to methane and carbon dioxide (Liu and Whitman, 2008). Although most methanogenic species are hydrogenotrophs, two-thirds of biologically produced methane is derived from acetate (Liu and Whitman, 2008). Combining isotopic composition of methane and the relation of methane to the sum of ethane and propane is an often-used method to distinguish natural gas from biogenic methane (Whiticar 2020). However, methane is oxidized by anaerobic and aerobic methanotrophs to carbon dioxide along its way to the atmosphere, which shifts the isotopic composition, adding more complexity. In case of organic rich soils or soils with a high groundwater table, methane production can outweigh its consumption leading to substantial methane emissions (Lai 2009). These processes are taking place in the active zone of the soil in general but especially environments with biogenic methane generation are prone to generate false positive well leakage classification."*

L50 – 60: Here you almost switch gears from oil and gas wells to a very detailed description of wetlands and peat bogs. I understand that some of the wells you examine here are located in such settings, but I feel like it would be beneficial to contextualize how common this is globally (i.e. for oil and gas wells to be located with peat bogs/wetlands at surface). Needs some grounding to show this is a typical environment that wells are hosted in and so is likely to be a common complexity – also to point out all soil systems whether they are wetland related or not are complex living, breathing systems and all have processes that might make evaluating leakage difficult without wider characterization and contextualization of processes... I think being more general, but using peat bogs as a particular case study, is a better general approach..

*We thank the reviewer for pointing this out and will rework the paragraph as mentioned above.*

L74-76: this is true for most/all settings – unless leakage is obvious! Which can often be the case in my experience (i.e. a leakage signal is massively overpowering and obvious to se with large fluxes and concentrations compared to reference sites). I think in general redrafting the intro to be more general and acknowledge the universal complexities would be a better way to frame this with the idea of peat bogs and their important in Europe/Germany (and elsewhere) properly

described/acknowledged...It's not clear currently how common it is for oil and gas wells to be hosted in such settings and so hard to gauge importance/significance...

> *We will redraft the introduction as recommended and highlight the frequency of abandoned wells in such a setting in Germany, which are about 15% of all wells.*

L80 – 83 – here you now address some of the above clarifications around how common this is (i.e. 15% of wells in Germany) but I feel this needs to come earlier (i.e. before the description of peat bogs etc) to justify why you focus on this so much....

> *After reworking the introduction, this information will be mentioned earlier as by the reviewer's recommendation.*

L83: "Such soils are highly likely to produce and emit methane" – this is true but perhaps a little too simplistic as how much is generated and how it is observed will also be variable depending on the seasons/soil conditions and other physical parameters – I feel this needs to be a bit better explained in the intro as per my main comment.

> *The reviewer is correct and we will put more emphasis on the parameters influencing the soil methane and its emission. However, as we are not focusing on seasonal and time depended effects of methane emissions in this study, we will only discuss this in the discussion part of the manuscript.*

L84-85 – Isn't a key focus also to determine the integrity status of the wells being examined? Seems like you should state that here.

> *As methane emission associated with abandoned wells is a sign for well integrity failure, the absence of methane emission is not a guarantee for intact well integrity. With our methodological approach, we are unable to directly investigate integrity failure in deeper parts of the well if leaking gas does not reach the upper 1 m of the soil. Analysis of deeper groundwater layers with regard to methane and easily soluble components like benzene would be better suited to assess integrity failure.*

L100 – 103: can you more clearly define the forest and meadow sites – some way to more rigorously quantify them perhaps with soil types or land use classification. Seems a little qualitative to state them this way – just wondering if you can be more accurate/robust and descriptive here

> *We will add more details to the site description. It will read:*
>
> *To investigate methane emissions related to abandoned onshore wells eight cut und buried wells in the south-eastern part of the oil field Steimbke-Nord covering an area of ~ 0.2 km² (Figure 1) were targeted. The eight wells are situated in three different land use types. Three wells (R-WA 272, R-WA 254, R-WA 264) are located in the western part of the area where active peat mining is ongoing with the bare peat directly at the surface (from here on "Peat" sites). Before the peat extraction in the active area began, the Peat site was also an*

*agricultural meadow that was probably temporarily grazed and regularly fertilized with manure like the meadow at well site R-WA 275, ~ 350 m to the east (from here on "Meadow" site). Two of the four wells from the Forest area (dominated by birch trees and pines) are located between the active Peat site and the Meadow (R-WA 273, R-WA 274), the remaining two in a larger forested area ~ 225 m to the north and northeast, respectively (from here on "Forest" sites).*

L106 – What do you mean "visible"? human eye visible? Likely the human eye cannot always capture all remnants of drilling. In my experience many drilling sites have altered and potentially contaminated soils (very old contamination perhaps not obvious) and so generally I would say that samples for lab analyses are needed to be more quantitative here. So, samples for TPHs, org c etc. and N to show the soil health and to compare this to the associated background locations. At the minute this is a very qualitative and loose statement. Can you be more quantitative?

> *We here meant human eye visible, as wall remains of the drilling cellar and other parts of the old well or nearby buildings/infrastructure were still in place. We will clarify this in the revised manuscript.*

Figure 1: Somewhat strange you provide results in the methods section and in figure 1 (i.e. methane sink or source) and seems like this occurs prematurely before you describe the methods for how measurements are taken etc. I suggest you need to remove these results and have figure 1 as a more generalized figure setting the scene for the whole study (without results included). I recommend a figure of all of Germany showing all wells and those which are in peat areas (i.e. the 15% you say that are) and then an inset of the area of interest showing no results and a combined inset of the generalized geological soil profile. Save these results for the results section..!!

> *We thank the reviewer for pointing this out and will remove the results from this figure. The recommended figure, however, would give the impression that the presented results are representative for the abandoned well situation in Germany or the Federal State Lower Saxony. As we do not want to convey this impression we therefore will stick to the setup of Figure 1 without results and will add the locations of soil gas sampling as "x" as additional information (see below).*

[Figure]

**Figure 1:** Overview of the study site in Steimbke with the well sites and reference site measuring grids each with 17 and 9 measuring points, respectively. Abandoned wells are depicted in white dots. The rough dimensions of the oilfield Steimbke-Nord are outlined by the yellow dotted line. Coordinates are stated in UTM-32U (WGS84) with easting and northing planar coordinates in meter. Blue indicates the well site measuring grid whereas yellow indicates the reference site measuring grids with the positions for soil gas sampling marked as white or black "x", respectively. The map was created using QGIS (v.3.22.3) and © Google Earth satellite images from 2015 as background.

Section 2.3: Flux measurements – did you determine an R2 value for the quality or stability of the flux measurements (i.e. how correlated with time the increase is – how smooth it is) – in my experience this is a very important parameter to evaluate and mention as it highlights how reliable the flux measurement is and potential methods of gas migration/source.... Also what about other weather parameters? Even if you didn't measure at the site can you get local weather station data to allow contextualization of the general weather patterns during measurements (e.g. temp, wind speed and barometric pressure cycles) – these are important to contextualize... and nice to mention and include...Important to show a single example of flux measurement data

*The smart chamber calculates the $r^2$ after the measurement and in our case, the majority of measurements had an $r^2$ above 0.5. From the measurements with $r^2$ <0.5 the vast majority were fluxes around 0 $CH_4$ flux with a range of ±0.1 nmol $m^{-2}$ $s^{-1}$. For a better understanding, we added three measurements below with fluxes of a) ~-0.187 nmol $m^{-2}$ $s^{-1}$, b) ~0.019 nmol $m^{-2}$ $s^{-1}$, c) ~121.66 nmol $m^{-2}$ $s^{-1}$. We will add this to the supplemental material.*

[Figure]

*Regarding weather parameters, we will add the parameters that we measured in the field with a handheld device (4200 Pocket Air Flow Tracker, Kestrel, Australia) to the supplements (Table S8). However, we do not plan to evaluate the fluxes with regards to barometric pumping as we consider that beyond the scope and design of our study with the temporal point measurements.*

Section 2.6: It is not immediately totally clear to me how oxidation rate is calculated – can you specify exactly how the rate is derived with the equation?

*Correct, we will add a description of how methane oxidation rates were calculated:*

*"In the end, methane oxidation was calculated as the slope of the declining methane concentration in µmol per incubation over time in a linear section of the graph. Subsequently, it was then accounted for the dry weight in case of MOx dry and the wet weight for MOx wet. Finally, to compare it to methane emissions MOx wet was multiplied by the respective soil density and a volume of 0.2 $m^3$, because 20 cm was the maximal depth of a composite sample."*

Section 3.1: This seems very much like methods to me and I think it needs moving into the methods section—it is setting the scene of the site/location investigated and not actually results. Please move such sections to the appropriate place...The results seem to start after L294....

*The reviewer is correct and we will move the information to section 2.1 and 2.2.*

L294 - 296 – Firstly are these 206 measurements from both wellhead locations **and** reference sites? Please clarify this. …

*The total amount of single measuring points is 206, for well and reference sites combined. We will clarify this in the revised version of the manuscript.*

…Next, I have some concerns with how this is described in general that makes it hard to follow and potentially a little misleading. It should be expected that soils emit and take up methane naturally and that typically in natural soils we get close to (but not typically exactly) net zero (i.e. +/- 0.05 µmol/m2/sec CH4 flux) – good to be clear on that as what you see falls within natural ranges – in my view you need to describe and contextualise natural ranges better in the article to help the reader understand what is normal and what might be anomalous....

*We agree with the reviewer and will add some natural ranges to the introduction and discuss these in more detail in the discussion.*

*The introduction will read: "To put this into perspective, upland forests are known to act as methane sink taking up to ~–4 nmol $CH_4$ $m^{-2}$ $s^{-1}$ methane from the atmosphere, whereas natural wetlands emit up to ~600 nmol $CH_4$ $m^{-2}$ $s^{-1}$, which can be topped by rice paddy fields with over 2000 nmol $CH_4$ $m^{-2}$ $s^{-1}$ (Oertel et al. 2016)."*

…Next, if this is all measurements (i.e. 206 from wellhead and background lumped together) you need to differentiate these and then compare well head and reference location by site – I find not differentiating makes describing the results in this way limited. Also to refer to the lowest flux as –ve is not really correct. A negative result is CH4 uptake, not a small flux which would still be positive.

*We will add box-whisker plots to visualize the differences between reference and well sites, for better visualization (see below). As described in the manuscript the box-whisker plots will underline that most well sites do not differ to their respective reference site. Except for well sites WA-274 and WA-254, however, in case WA-254 the reference site showed higher emissions, and in case WA-274 was a stronger methane sink. For the statistical tests, we used the Kruskal-Wallis-Test to test for normal distribution, which the methane fluxes did not show. The Mann-Whitney-U-test was then deployed to compare well and reference site data. In addition, we will change the wording to "methane uptake".*

[Figure]

**Figure 6:** Box-whisker plots for the measured well and reference sites: (a, b, c, d) forest, (e) meadow, (f, g, h) peat. Outliers are displayed as points. The underlying statistical parameters are listed in Table S9.

L298 – 300: These are all within typical natural ranges to my knowledge (i.e. are not at all indicative of anomalous processes or leakage) – so you need to better describe what natural ranges are and then state these are within them...

*We agree with the reviewer, however, in our opinion the result part is not the right place for this. As described above we will cover this in the introduction and discussion part of the manuscript.*

L301 – 305: Again – these all seem to be within natural ranges for such settings – you need to state that and better articulate typical natural ranges in the paper.

*Again, we thank the reviewer and agree that this aspect is very important. As mentioned above we will cover this in the introduction and discussion.*

L306-312: The >500nmol value seems higher and potentially approaching the outer boundary of typical natural ranges – again you need to define better what are potentially natural ranges for peat bogs/similar settings. Here as well you refer to "high" fluxes – but what is this relative to? – …

*That is right, but it is still in the natural ranges for peat bogs: However, we will point out that this is high compared to the other fluxes we measured but still low in comparison with fluxes measured at many leaking abandoned (Bowman et al. 2023, Cahill et al. 2023).*

…this value is still quite low compared to other reported leakages at oil and gas wells (see Cahill et al Evaluating methane emissions from decommissioned unconventional petroleum wells in British Columbia, Canada for example). In fact, this also highlights a lack of review or identification of the types of rates that are observed when leakage is happening from a well. In general I think you need to better review and state this in the intro and also then refer to it here when comparing your results…

*We agree with the reviewer and will add more context in the introduction and especially in the discussion on emissions from abandoned wells and natural ranges.*

… Other good papers to mention/review that are very similar to what you do here (but are not acknowledged) are Samano et al Constraining well integrity and propensity for fugitive gas migration in surficial soils at onshore decommissioned oil and gas well sites in England and Forde et al Identification, spatial extent and distribution of fugitive gas migration on the well pad scale – among many others. I think what you see here is not indicative of leakage but to say that with confidence you need to robustly describe what leakage looks like as well as natural ranges...

*We agree with the reviewer and will point this out in the discussion of the manuscript. The reference Samano et al. (2022) will be added.*

Table 2 and figure 3 would benefit from inclusion of typical natural ranges for the soil/land types and also perhaps inclusion of reported values for leaking oil and gas wells to show how these values compare......

*We will add natural ranges to Figure 3 as well as Table 2, and leaking well fluxes to Table 2.*

[Figure]

**Figure 4:** Methane flux (a, d, g), soil gas methane concentration (b, e, h), and potential methane oxidation rates (MOx; c, f, i) depicted as histograms for well (blue) and reference sites (yellow) at the three areas forest (a, b, c), meadow (d, e, f), and peat extraction site (g, h, i). The red line in a, d, g indicates zero flux, sites left of the line acted as net methane sinks and at the right as net methane sources. The grey background represents natural ranges mentioned in literature (Abdalla et al. 2016, Oertel et al. 2016).

Figure 4 – there appears to be no differentiation between reference sites and well heads in this figure – suggest to show that for clarity….

> *We had the same idea as the reviewer, however, we dismissed it because we believe that the additional information do not add enough value and reduces the clarity of the illustration. Here is part of the figure to underline this.*

[Figure]

**Figure 2:** Depth profiles of (a) $CH_4$, (b) $CO_2$ soil gas concentrations, as well as (c) $\delta^{13}C\text{-}CH_4$ and (d) $\delta^{13}C\text{-}CO_2$ values for Forest (brown diamonds), Meadow (light green squares) and Peat (dark grey circles) sites, the respective reference sites are displayed as empty symbols. Note the logarithmic scales in (a) and (b). Isotopic composition of (c) methane and (d) carbon dioxide is depicted as difference to the Vienna Pee Dee Belemnite (VPDB) standard.

… Also again seems like have some indicators of natural ranges (e.g. atm concentration also) on this figure would allow the reader to better judge how the observed results deviate from that.

> *We agree with the reviewer and will add the respective atmospheric values at 0 cm (surface) for clarification.*

General comment on the results – I am quite interested to see the high levels of CH4 in soils in the meadow but guess this is a result of the underlying Peat? Overall seems like this strong natural CH4 signature would make identifying leakage much more difficult than other settings. I also wonder what the mixture of potential thermogenic gas and natural peat derived gases would look like. It would be beneficial to create a simple mixing line to show this and then plot the observed gas concentration on the same plot to see if and where they fall in relation to this mixing line. Can this be done?

> *That is right, the isotopic data and absence of higher hydrocarbons suggest, that the methane was produced microbially in the underlying peat, and that this is making the potential methane emissions from leaking wells challenging. We assume a similar peat profile as displayed in Figure 2d, with topsoil and grass on top. As for the suggested mixing*

*line, we are missing the thermogenic endmember and have multiple biogenic endmembers, one for each site, thus were not able to provide an adequate mixing scenario. Such mixing scenarios in diagrams must also be used with great caution and can be misleading, since microbial oxidation, which our study suggests is the cause of various $\delta^{13}C$ data, shifts data in isotope diagrams (towards potentially more thermogenic end members).*

Figure 5: really nice microbial data. It seems you do not delineate between reference sites and well heads in this figure --- is this correct? So, this data is for both reference and wellhead locations? Seems like it would be useful to show somehow this delineation in the results.

*These are indeed samples from both well and reference sites. However, as the soil gas flux data did not indicate any leakage we combined these samples for this analysis. In addition, all used samples and their respective methanotrophic community are listed in detail in Table S6 in the data supplement.*

Section 4.1: A good overall discussion. However, would point out a very recent study showed also the extent of methane oxidation that can occur in soils around leaking wells (in Canada and where it was also cut and buried) this study would be useful to mention as it supports everything you are saying here (Cahill et al. Evaluating methane emissions from decommissioned unconventional petroleum wells in British Columbia, Canada)….

*We thank the reviewer for pointing this out and will add a comparison with the mentioned article in a new paragraph in section 4.2 of the discussion.*

… As for factors affecting spatiotemporal variability – I think this discussion could be improved with more references that demonstrate these factors. For example, Forde et al "Barometric-pumping controls fugitive gas emissions from a vadose zone natural gas release" showed how barometric pumping controls surface manifestation of leakage and fluxes very robustly and this seems like a paper worth mentioning….

*We agree with the reviewer and will add the respective reference.*

… Was it not possible to get weather station data to evaluate against your results in this study?

*As mentioned above, we will add the snapshot data from our handheld device to the supplement (Table S8) but do not want to discuss this in detail in the manuscript, as the influence of such parameters on the emissions were not the focus of our study. We will describe this important effect, however.*

L630 – 632- I think some would disagree with this statement. For example, British Columbia Canada actually has very good regulations and most development occurred since the 1960s (with most since 2000) – so in that case they have mostly plugged and abandoned wells.

*We thank the reviewer for pointing this out and will change the wording so it fits this information.*

L644 -648 – Again this was also thoroughly examined in Cahill et al "Evaluating methane emissions from decommissioned unconventional petroleum wells in British Columbia, Canada". It might be worth comparing your numbers to those observed in that study….

*We will add the mentioned reference to the discussion and a comparison of the data as mentioned above.*

---

## Author Comment (AC2)

**Response to comments by referee 2**

The authors present a field and lab study on the biogeochemistry of methane in peat soils as related to potential leakage of deep, abandoned oil wells. The study is fine with respect to the field and lab measurements done and the authors show they are at home in the topic of biogeochemistry of methane. The peculiar aspect of the manuscript is its framing. The authors refer to the need to know background emissions from the shallow subsurface when it comes to potential leakage from the deeper subsurface. I fully agree with this need but several remarks must be made:

> *We would like to thank the reviewer for the very thorough review and the positive general assessment of our study and results.*

1. the title is misleading as it is very general. It should be indicated in the title that this study deals with peat soils as they have the highest potential for methane emissions (probably together with paddy rice fields)

> *The reviewer is right to point this out, we will change the title to better fit the presented work:*
>
> *"Interferences caused by the biogeochemical methane cycle in peats during the assessment of abandoned oil wells"*

2. the authors were unfortunate that no thermogenic methane was found at all. However, they stick to searching for thermogenic methane which makes the manuscript in a way forced in its scope. Section 4.2 is a peculiar read as well as other parts of the manuscript.

> *The reviewer is correct in the impression that the focus of the study is the evaluation of potentially leaking oil wells and possible complications in peat with high methane cycling. We were not unfortunate to not find any thermogenic methane, but we do consider the very strong interpretation of our data to be necessary. We agree with the reviewer that section 4.2 has to be reworked. We will shorten the comparison with the soil gas approach by Romanak et al. 2012 and incorporate a comparison with natural methane fluxes and other studies that found methane emissions from leaking abandoned wells.*

3. the study is relevant when it comes to greenhouse gas emissions and climate change. This comes around the corner at a rather late stage in the manuscript. The scope of section 4.3 should be introduced much earlier.

> *We thank the reviewer to point this out and will rework part of the abstract and introduction to better highlight this part of the scope.*

4. the authors studied peat soils above abandoned oil wells. This is a somewhat poor choice since oil reservoirs often have lower liquid pressures than the surrounding rocks due to the exploitation for oil. Hence, there is oil and/or water flow from surrounding rocks to the reservoir (often not completely depleted). Depending on the exact composition of the field, there may be more or less natural gas involved in oil reservoirs which has a tendency to move upwards because of buoyancy effects but this may be restricted depending on the resulting pressure field. It would have been much, much better when the researcher had studied peat soils above abandoned gas wells. It seems to me that the authors are not aware of this major difference. They however should indicate this essential difference. Here, I realise that local geology plays a role in the way reservoirs may be connected or not to above as well as potential leakage of gas or oil along or through (abandoned) wells.

*The reviewer points out an important potential difference between oil and gas wells, which the community and we are fully aware of, however earlier studies showed substantial methane emissions from plugged/abandoned oil wells (e.g., $4.6 \times 10^{-2}$ g $h^{-1}$ – 0.13 g $h^{-1}$, Williams et al. 2021). These emissions were lower than emissions from gas wells ($4.1 \times 10^{-3}$ g $h^{-1}$ – 18 g $h^{-1}$, Williams et al. 2021) but still substantial. Methane emissions from oil wells were also reported in other studies (Kang et al. 2014, Townsend-Small et al. 2014, Saint-Vincent et al. 2020).*

*Furthermore, oil wells in Germany are typically older than gas wells, as gas exploration started in the 1960s, and oil wells (which started in ~1860 and prospered after 1900) are thus more prone to integrity failure as material and technology were less advanced. Additionally, due to the higher number of oil fields in Germany and the production dependent higher number of wells in an oil field in general, there are 6 times more oil wells than gas wells in Germany.*

*We want to point out, that this was a pilot study, with the aim to test our methodology particularly in an area with abundant biogenic methane cycling. We did not aim nor wanted to convey the impression that this study is representative for abandoned oil and gas wells in Germany, to make this clearer we omitted the "gas wells" from the title. In addition, we are currently working on an additional study to address the abandoned well situation in Germany for which we already measured and are still planning to sample multiple gas wells and more oil wells as well.*

Especially based on remarks 3 and 4, I suggest that the authors rewrite the manuscript where they present a new framework addressing both greenhouse gas emissions from peat lands and dealing with shallow biogenic methane emission when addressing potential leakage of thermogenic methane. If not, I must recommend to reject the manuscript.

*We addressed these points and hope that especially our answer to remark 4 was able to convince the reviewer, that it is important to consider oil wells as well. In addition, our study design was established to address the question if our methodology was sufficient to detect methane emissions in the vicinity of abandoned wells and more importantly if we are able to pin point the source of such methane in a complex and organic rich setting. Another framework, for example focusing on greenhouse gas emissions from peat lands as suggested, is beyond the scope of this study and its design. This would result in shortcomings regarding the sampled parameters and for example lack of seasonality. With this in mind, we carefully balanced the selected scope and hope the*

*reviewer will approve the changes to the manuscript and the resulting clearer message we want to convey.*

*We further want to point out, that we did not sample pristine peatlands but sampled three anthropogenically influenced peat-rich areas that had been drained decades or even centuries ago.*

In addition, I have the following general remarks in addition to individual remarks annotated in the manuscript:

1. carefully check on the use of single versus plural. Subjects and related verbs are frequently not in harmony.

*We thank the reviewer for pointing this out and will follow the thorough individual remarks in the uploaded pdf and we will additionally check the manuscript carefully again regarding these points.*

2. the RESULTS become boring to read when presenting all kinds of numbers. These numbers should be presented in Tables (as done) and patterns should be discussed in terms of higher, lower, etc. Very importantly, it remains unclear whether the different sites differ statistically significant or not. I recommend that significance tests get added and box-whisker plots get presented (instead of means or so in tables that may be moved to supplementary material).

*Indeed, the results included a lot of numbers, we will revise the marked parts and will include box-whisker plots in the result section (see below). For the statistical tests we used the Kruskal-Wallis-Test to test for normal distribution, which the methane fluxes did not show. The Mann-Whitnes-U-test was then used to compare well and reference site data. For WA-211, WA-209, WA-273, WA-264, WA-272, WA-275 well and reference sites were similar with regard to methane fluxes. WA-274 and WA-254 however showed significant differences in fluxes between well and reference sites, however, in case WA-254 the reference site showed higher emissions, and in case of WA-274 the well site a lower methane sink.*

*We further plan to stream-line the result section and focus more on trends rather than single numbers.*

[Figure]

**Figure 1:** Box-whisker plots for the measured well and reference sites: (a, b, c, d) forest, (e) meadow, (f, g, h) peat. The underlying statistical parameters are listed in Table S9.

3. the DISCUSSION presents a lot of results which should be avoided. Move more effectively to the discussion topics.

> *We plan to restructure the discussion section and implement a clearer separation of results and discussion. However, in our opinion is the interpretation of data, as for example using the isotopic plots of figure 7 part of the discussion.*

4. the CONCLUSIONS are poorly written. It reads more like a summary than conclusions

> *We will revise the conclusions to underline the implications of our study better. The mentioned comparison with Bowman et al. 2022 from the original conclusion will be moved to the discussion section 4.2. The conclusion will read:*

> *"While it is well known that abandoned oil and gas wells can have integrity issues, respective knowledge particularly on the 20,000 cut and buried wells in Germany, is lacking. Here we provide with our multi-methodological approach first data on potential methane fluxes from abandoned oil wells to the atmosphere. We combined*

*emission data (positive and negative) at wells and reference areas with gas geochemical characterization of soil gas samples to investigate eight wells in a peat rich setting with three different land use types (Forest, Meadow, Peat extraction).*

*The Peat extraction site was the only one, which emitted substantial amounts of methane. However, in general no difference in surface methane emission rates between well and reference sites independent of site characteristics (active peat mining, drained peat vegetated with birch trees or grassland) were observed. With respect to soil gases, the three areas showed highly variable but spatially correlating (i.e. area specific) methane concentrations concurring with $CO_2$ concentrations. The in-depth gas and isotope geochemical analysis revealed biogenic methane as source for the net emissions at the open peat site (methyl fermentation) and the meadow ($CO_2$ reduction pathway with partial methane oxidation). These findings and the absence of higher hydrocarbons excludes thermogenic gas emissions from the plugged wells.*

*Furthermore, the laboratory methane oxidation rates derived from our incubated peat samples demonstrated the capacity to counterbalance reported leakage rates for buried abandoned wells in other regions. The activity of such a microbial methane filter poses the risk for false negative leakage classification. Overall the observed methanotrophy could be highly relevant for Germany as 15% of our cut and buried wells are located in areas with very organic-rich soils. However, for a comprehensive evaluation of the situation of abandoned wells in Germany further investigations are needed. Therefore, additional sampling at different sites (oil/gas wells of different age and deconstruction histories) in Northern Germany with the here introduced methodology are under way and we will evaluate the natural mitigation potential for different soil types and land uses.*

*In conclusion, exclusively using emission-based approaches are not suited for integrity failure assessments of buried wells as these would be susceptible to misinterpretations. We highly recommend a holistic approach for surveillance including the determination of methane emission, soil gas composition and isotopic signatures at and in the vicinity of well sites against the background of a carefully selected reference site."*